Molecular Biology and Physiology

# *Dinoroseobacter shibae* Outer Membrane Vesicles Are Enriched for the Chromosome Dimer Resolution Site *dif*

Hui Wang,[a] Nicole Beier,[a,b] Christian Boedeker,[c] Helena Sztajer,[a] Petra Henke,[c] Meina Neumann-Schaal,[d] Johannes Mansky,[a] Manfred Rohde,[e] Jörg Overmann,[c] Jörn Petersen,[c] Frank Klawonn,[f,g] Martin Kucklick,[a,b] Susanne Engelmann,[a,b] Jürgen Tomasch,[h] Irene Wagner-Döbler[a]

[a]Institute of Microbiology, Technical University of Braunschweig, Braunschweig, Germany
[b]Research Group Microbial Proteomics, Helmholtz Centre for Infection Research (HZI), Braunschweig, Germany
[c]Department of Microbial Ecology and Diversity Research, Leibniz Institute DSMZ—Deutsche Sammlung von Mikroorganismen und Zellkulturen, Braunschweig, Germany
[d]Junior Research Group Bacterial Metabolomics, Leibniz Institute DSMZ—Deutsche Sammlung von Mikroorganismen und Zellkulturen, Braunschweig, Germany
[e]Central Facility for Microscopy, Helmholtz Centre for Infection Research (HZI), Braunschweig, Germany
[f]Bioinformatics and Statistics Research Group, Department of Cellular Proteomics, Helmholtz Centre for Infection Research (HZI), Braunschweig, Germany
[g]Department of Computer Science, Ostfalia University of Applied Sciences, Wolfenbüttel, Germany
[h]Department of Molecular Bacteriology, Helmholtz Centre for Infection Research (HZI), Braunschweig, Germany

Hui Wang, Nicole Beier, Christian Boedeker, and Helena Sztajer contributed equally to the study. The order of names was decided based on experimental and intellectual contribution to the study.

**ABSTRACT** Outer membrane vesicles (OMVs) are universally produced by prokaryotes and play important roles in symbiotic and pathogenic interactions. They often contain DNA, but a mechanism for its incorporation is lacking. Here, we show that *Dinoroseobacter shibae*, a dinoflagellate symbiont, constitutively secretes OMVs containing DNA. Time-lapse microscopy captured instances of multiple OMV production at the septum during cell division. DNA from the vesicle lumen was up to 22-fold enriched for the region around the terminus of replication (*ter*). The peak of coverage was located at *dif*, a conserved 28-bp palindromic sequence required for binding of the site-specific tyrosine recombinases XerC/XerD. These enzymes are activated at the last stage of cell division immediately prior to septum formation when they are bound by the divisome protein FtsK. We suggest that overreplicated regions around the terminus have been repaired by the FtsK-*dif*-XerC/XerD molecular machinery. The vesicle proteome was clearly dominated by outer membrane and periplasmic proteins. Some of the most abundant vesicle membrane proteins were predicted to be required for direct interaction with peptidoglycan during cell division (LysM, Tol-Pal, SpoI, lytic murein transglycosylase). OMVs were 15-fold enriched for the saturated fatty acid 16:00. We hypothesize that constitutive OMV secretion in *D. shibae* is coupled to cell division. The footprint of the FtsK-*dif*-XerC/XerD molecular machinery suggests a novel potentially highly conserved route for incorporation of DNA into OMVs. Clearing the division site from small DNA fragments might be an important function of vesicles produced during exponential growth under optimal conditions.

**IMPORTANCE** Gram-negative bacteria continually form vesicles from their outer membrane (outer membrane vesicles [OMVs]) during normal growth. OMVs frequently contain DNA, and it is unclear how DNA can be shuffled from the cytoplasm to the OMVs. We studied OMV cargo in *Dinoroseobacter shibae*, a symbiont of dinoflagellates, using microscopy and a multi-omics approach. We found that vesicles formed during undisturbed exponential growth contain DNA which is enriched for genes around the replication terminus, specifically, the binding site for an enzyme complex that is activated at the last stage of cell division. We suggest that the enriched genes

Address correspondence to Jürgen Tomasch, Juergen.Tomasch@helmholtz-hzi.de, or Irene Wagner-Döbler, I.Wagner-Doebler@tu-bs.de.

Outer membrane vesicles have been studied in Dinoroseobacter shibae, a marine bacterium. DNA and protein cargo suggest that these vesicles are produced at the last stage of cell division and might clear the divisome from waste DNA.

are the result of overreplication which is repaired by their excision and excretion via membrane vesicles to clear the divisome from waste DNA.

**KEYWORDS** DNA repair, OMV, vesicles, replication termination, circular chromosomes

The formation of membrane vesicles and their release into the extracellular environment is a fundamental trait of cells from all domains of life (1), and it has even been suggested to represent the mechanism for the evolution of the endomembrane system of eukaryotic cells (2). All Gram-negative bacteria form membrane vesicles from the outer membrane (OMV) in a process called blebbing or budding (3), triggered by a local reduction of covalent cross-links between the outer membrane and the peptido-glycan layer, often in microdomains which are enriched with secondary metabolites, specific glycolipids, or misfolded proteins (4). The size, lipid composition, and cargo of OMVs depend strongly on environmental parameters and growth conditions, and accordingly, they can adopt different functions in the same species (5). In *Salmonella*, the shedding of OMVs accelerates the remodeling of the lipopolysaccharide (LPS) composition of the outer membrane which is an important adaptation to environmental transitions (6). DNA and RNA are often components of OMVs (5), which might therefore potentially represent a new mechanism for horizontal gene transfer (7). The periplasmic space is free of DNA; thus, it is unclear how DNA can be incorporated into OMVs. Moreover, while differences in OMV structure and cargo are clearly controlled by environmental transitions, OMVs are also secreted under stable conditions during normal growth by all bacteria. Thus, a conserved mechanism for their biogenesis would be expected.

In the ocean, OMVs were discovered to be freely suspended (8). They have approximately the same abundance as bacteria, a distinct depth distribution, and contain DNA from a variety of marine bacterial taxa (9). OMVs of the most abundant marine phototrophic bacterium, *Prochlorococcus*, were studied in laboratory culture and could be "infected" by a cyanophage and, in such a way, might protect life cells from phage attack (8). OMVs from *Vibrio* spp. play important roles for pathogenesis and symbiosis: they act as ferries for hydrolytic enzymes and signaling molecules in the coral pathogen *Vibrio shilonii* (10) and the oyster pathogen *Vibrio tasmaniensis* (11). They can package the hydrophobic quorum sensing signal CAI-1 from the human pathogen *Vibrio harveyi* (12). In the symbiotic relationship of *Vibrio fischeri* with the bobtail squid, OMVs induce host differentiation in a pH-dependent way (13, 14). *Vibrio* spp. have sheathed flagella, and flagella rotation increased the amount of secreted OMVs in *V. fischeri* (13). Strikingly, OMVs might play a key role in the symbiotic relationship between the flatworm *Paracatenula*, which has neither mouth nor gut, and its obligate chemosynthetic symbiont "Candidatus *Riegeria santandreae*." They might deliver energy-rich compounds to the host tissue and thus protect the symbiotic bacteria from being digested (15).

Here, we analyzed OMVs from a model strain of the roseobacter group. The roseobacters are marine representatives of the *Rhodobacteraceae*, a family of proteobacteria within the order *Rhodobacterales* in the class *Alphaproteobacteria* (16), and can be very abundant in coastal areas, algae blooms, and the polar oceans (17). *Dinoroseobacter shibae* has been isolated from the dinoflagellate *Prorocentrum lima* (18) and can supply marine algae with B vitamins for which they are auxotrophic (19) but kills the algae at later growth stages (20). Its genome is composed of one chromosome, three plasmids, and two chromids (21). Two of the plasmids carry type 4 secretion systems and can be conjugated into distantly related roseobacters (22). The *D. shibae* chromosome encodes the gene transfer agent (GTA); these phage-like particles are inherited vertically, but they transfer fragments of host DNA horizontally within the population (23). In *D. shibae*, GTA synthesis is quorum-sensing controlled and suppressed in the wild type by the product of the autoinducer synthase LuxI$_2$ (24).

We investigated OMVs of *D. shibae* during undisturbed growth in defined minimal medium. We determined their abundance, size, ultrastructure, and DNA content and

observed their biogenesis *in vivo* by time-lapse microscopy. DNA from the OMV lumen was sequenced, and its coverage relative to chromosome, plasmids, and chromids was calculated. The fatty acid composition of vesicle membranes was compared to that of cell membranes. We performed an extensive proteome analysis of four fractions: (i) membranes of vesicles, (ii) membranes of whole cells, (iii) soluble proteins of vesicles, and (iv) soluble proteins of whole cells. Our data led us to formulate a model for the incorporation of DNA into OMVs of *D. shibae* based on the key observation that the recognition sequence of the FtsK-XerC/XerD machinery as well as the region of the hypothetical terminus region of the chromosome is highly enriched in the enclosed DNA.

(This article was submitted to the online preprint archive [25].)

## RESULTS AND DISCUSSION

**Microscopic observations show constitutive secretion of outer membrane vesicles containing DNA in *Dinoroseobacter shibae*.** Since little is known about vesicles excreted by roseobacters, we first studied their formation and ultrastructure in *D. shibae* during undisturbed growth in minimal medium. The average number of vesicles released per cell was determined in the culture supernatant directly by NanoSight particle tracking analysis, and bacterial cells were counted by flow cytometry (see Fig. S1 in the supplemental material). Vesicle numbers increased continually during growth and peaked at the end of the exponential growth phase. The ratio of vesicles per cell fluctuated around 0.75, ranging from 0.4 to 1.2. Thus, they were produced constitutively throughout growth. We concentrated and purified vesicles from the supernatant of an exponentially growing culture and determined size and abundance using transmission electron microscopy (TEM) and particle tracking analysis. TEM analysis of the distinct band visible after ultracentrifugation showed densely packed vesicles (Fig. 1A). Their diameter ranged mainly from 20 nm to 75 nm (Fig. 1B); rarely, larger vesicles up to 210 nm were found. The particle tracking analysis with NanoSight also revealed a broad size range of the vesicles (Fig. 1B). Maxima were detected at 68, 76, 107, 130, 210, and 282 nm. The average size of the vesicles was 53 nm based on TEM pictures and 93 nm based on NanoSight data (Fig. 1B). The larger average value resulting from NanoSight data is most likely caused by the detection limit of the NanoSight instrument, which cannot track vesicles <50 nm, which are visible on the TEM images.

We then studied vesicles of *D. shibae* on the single-cell level. Using TEM of thin-sectioned cells, we were able to capture the biogenesis of vesicles. They were linked with a thin membrane sleeve to the releasing cell and were composed of an outer membrane (Fig. 1C, black arrows). The cytoplasmic membrane was not part of the vesicles (Fig. 1C, white arrows). Additional examples of vesicle formation from the outer membrane are shown in Fig. S2. No double membrane vesicles were observed. Thus, we will subsequently term the vesicles excreted by *D. shibae* outer membrane vesicles (OMVs).

Time-lapse microscopy captured examples of OMV secretion from single cells. In the first example (Fig. 1D; see also Movie S1), OMVs were secreted one after the other at the division plane over a period of 13 h and then moved away from the division plane but stayed close to the releasing cell. Cell division and cell growth appeared to be halted. Only after OMVs had been released, cell division was completed and cells continued to grow. In the second example (Fig. 1E; see Movie S2), a single OMV appeared at the division plane, detached from the cell, and was released to the supernatant within 1 to 2 h.

OMVs of Gram-negative bacteria have often been observed to contain DNA both in their lumen and on the surface (e.g., see reference 26). To differentiate between intra- and extravesicle DNA, we stained OMVs with the two fluorescent dyes 4',6-diamidino-2-phenylindole (DAPI) and *N*-(3-triethylammoniumpropyl)-4-[4-(dibutylamino)styryl] pyridinium dibromide (FM1-43) (Fig. 1F to H). While DAPI can penetrate membranes, FM1-43 only emits a signal when it is inserted into a membrane; thus, unspecific background staining is avoided. A fraction of the concentrated OMVs visible under the phase-contrast microscope (Fig. 1F) were stained with FM1-43, indicating intact

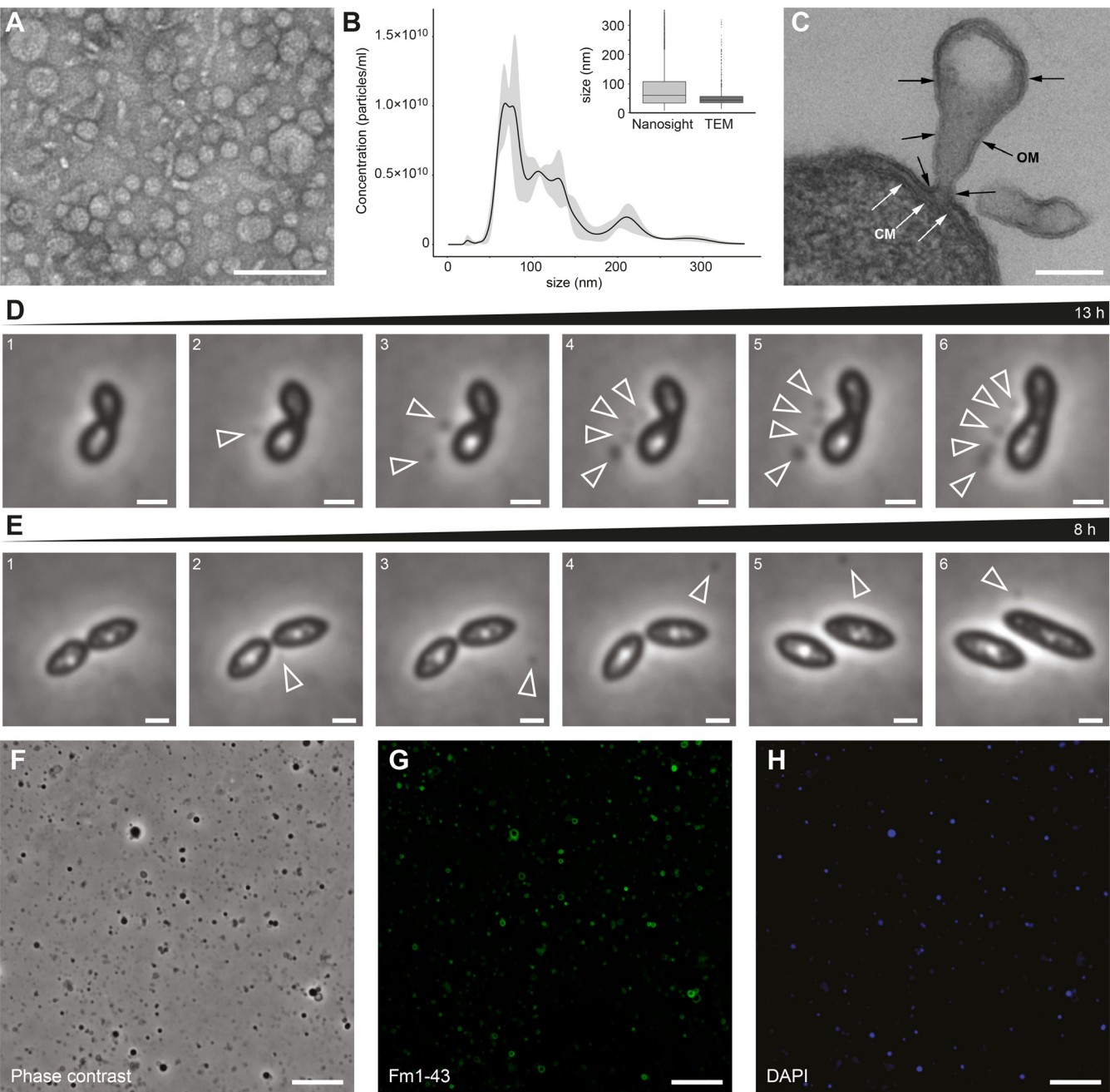

**FIG 1** Vesicle formation in the *D. shibae* cell population. (A) Transmission electron microscopy (TEM) showing negatively stained vesicles purified by ultracentrifugation from an exponentially growing culture of *D. shibae*. (B) Size and abundance of vesicles determined by NanoSight particle tracking analysis. (Inset) Comparison of NanoSight and TEM quantification; $n$ = 2,406 and 1,421, respectively. (C) TEM of ultrathin sections shows OMVs originating from the outer membrane. Scale bars in panels A and C, 200 nm. (D and E) Two instances of vesicle formation observed by phase-contrast time-lapse microscopy. Representative images are shown here, see also Movies S1 and S2 in the supplemental material. (D) Vesicle formation was followed over 13 h. An increasing number of OMVs were released from the division plane of the same cell over time (arrowheads). The vesicles stayed attached to the donor cell. The OMV-releasing cell stopped dividing during vesicle segregation. (E) Vesicle formation was followed over 8 h. A single OMV was formed and released into the supernatant. The OMV appeared at the division plane and started to move around the cell. Scale bars, 1 μm. (F to H) Light microscopic detection of DNA within OMVs. Phase contrast (F), membrane staining with FM1-43 (G), and DNA staining with DAPI (H) are shown. Scale bars in panels F to H, 10 μm.

membranes (Fig. 1G), or DAPI, indicating DNA (Fig. 1H). OMVs stained with both DAPI and FM1-43 were counted to estimate the percentage of DNA carrying OMVs. By analyzing 10 fields of view in two different experiments ($n$ = 20,349), we found that approximately 65% of the vesicles carried DNA that was detectable by fluorescence staining.

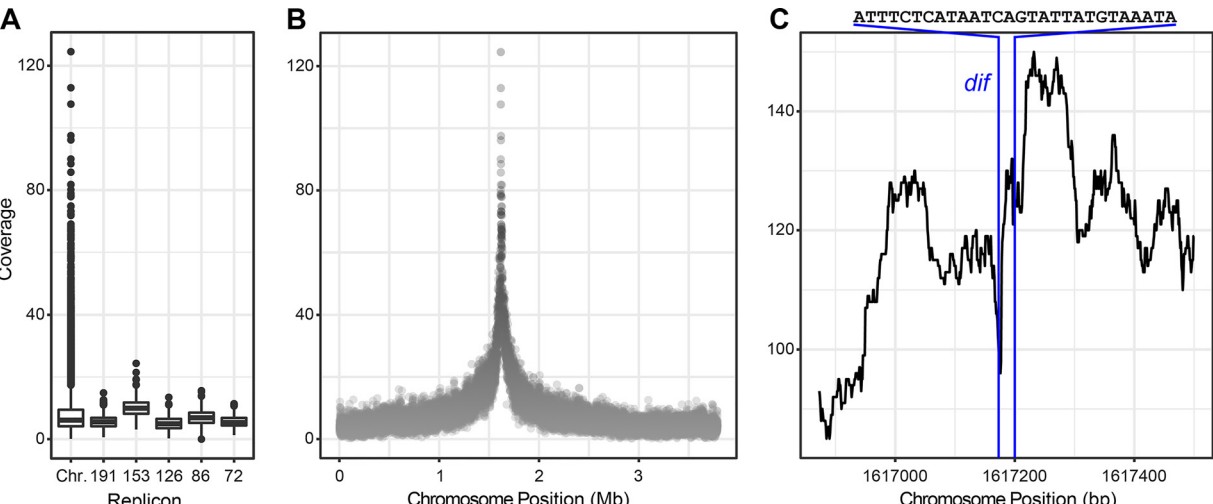

**FIG 2** DNA from OMVs of *D. shibae* was enriched for the region around the terminus of replication (*ter*). (A) Sequence coverage of chromosome and plasmids in DNA inside OMVs. Median, minimum, and maximum values are shown. Plasmids are abbreviated according to their size: 191, 152, 126, 86 and 72 kb. See Fig. S3 for two additional biological replicates. (B) Sequence coverage of the chromosome showing overrepresentation of the region on both sides of position ~1.6 Mb, which is the region around the replication terminus. Coverage was calculated for a sliding window of 500 nt in panels A and B. See Data Set S1, sheet 3, for an identification of the enriched genes at coverages of >40-fold and >100-fold. (C) Sequence coverage of the region around chromosome position 1.6 Mb at single-base resolution. The position and 28-bp nucleotide sequence of *dif* are indicated.

Very few cells in the population produce OMVs. This is shown by the scanning electron micrographs (SEM) of an exponentially growing culture; only few cells have vesicles on their surface (see Fig. S2C), although many cells have little bumps on their surface which could be emerging vesicles. A closer look reveals that cell size varied strongly in the population, as *D. shibae* is characterized by morphological heterogeneity, which is under the control of quorum sensing (27). Figure S2C shows clusters of cells with different morphologies and different numbers of vesicles. While some cells lacked OMVs completely, others were covered with many small and a few larger OMVs. OMV size was measured in the SEM picture and was between 26 and 75 nm.

**The chromosomal region around the terminus of replication is strongly enriched in *D. shibae* OMVs.** Vesicles could represent a new mechanism for gene transfer (28). To determine if the complete genome might potentially be horizontally transferred through OMVs, we sequenced the DNA from OMVs of *D. shibae*. DNA that might be attached to the vesicle surface or was released from disintegrating cells during vesicle preparation was removed by treating the concentrated vesicle preparation with DNase. As a control, we sequenced the total DNA from the cell-free purified vesicles without DNase treatment, containing extravesicle DNA, DNA attached to the vesicles, and the DNA contained inside the vesicles.

Figure 2 shows one of the three biological replicates of the DNase-treated samples (all three biological replicates are shown in Fig. S3A). The sequenced reads covered the complete chromosome as well as all five extrachromosomal replicons, albeit with a low median coverage between 5- to 13-fold (Fig. 2A; see also Fig. S3B and Data Set S1, sheets 1 and 2). However, a specific region of the chromosome around 1.6 Mb was strongly enriched in vesicle DNA (Fig. 2B and S3B). The three biological replicates showed consistent results, with a maximum coverage at this peak ranging between 100- and 400-fold (Fig. S3A and B), which represents an enrichment of 12- to 22-fold compared to the median coverage (Fig. S3B). In samples that were not treated by DNase, the coverage of extrachromosomal elements was roughly according to their copy number in the cell (24) (Fig. S3B). Chromosomal reads from the untreated control also showed a peak around 1.6 Mb. However, the coverage declined in a linear way compared to the steep decline in the DNase-treated samples (Fig. S3C). The peak DNA was not strongly enriched compared to the median coverage (Fig. S3B). Of note, the

coverage patterns for both DNase-treated and control samples observed here are strikingly different from the homogeneous coverage regularly observed when sequencing genomic DNA. A strong overrepresentation of specific chromosomal regions was previously found for the DNA packaged into gene transfer agent particles (24), which was completely unlike the coverage pattern observed here.

We conclude that a large amount of genomic DNA was present in the untreated extract, probably derived from disrupted cells and attached to the exterior of the vesicles. Inside of the vesicles, DNA fragments covering the complete genome of *D. shibae* were also found, albeit at a very low coverage. The chromosomal region around 1.6 Mb was strongly enriched in the DNase-treated vesicles only. This region is located around the probable terminus of replication (*ter*) in *D. shibae* (29). Using 40-fold coverage as a cutoff, we analyzed the genes contained in this enriched chromosomal region and found that *cckA*, *recA*, *ctrA*, and *gafA* were among them as expected (Data Set S1, sheet 3) (29). The enriched genes spanned a continuous region of 170 genes on the chromosome, from Dshi_1477 to Dshi_1647. Using a 100-fold coverage as cutoff, we found 10 genes, which again spanned a continuous region on the chromosome from Dshi_1554 to Dshi_1563 (Data Set S1, sheet 3).

**The *dif* sequence is present in the most highly enriched OMV DNA.** We then studied the enriched genomic region within the OMVs at single-base resolution (Fig. 1C; Data Set S1, sheets 1 and 2). Strikingly, the peak of coverage at 1.6 Mbp was resolved into three peaks, of which the central one corresponds exactly to the *dif* (<u>d</u>eletion <u>i</u>nduced <u>f</u>ilamentation) sequence of *D. shibae*. The role of this palindromic 28-bp sequence during replication has been studied in such detail that it is possible to develop a hypothesis how DNA fragments containing the *dif* region can arise and be exported into OMVs. This hypothesis will be developed below and is schematically shown in Fig. 3.

The *dif* sequence is located opposite the origin of replication (*ori*) close to the terminus (*ter*) region on the circular chromosomes of bacteria (30). It represents the binding site for two site-specific recombinases, XerC and XerD (31). The function of the XerC/XerD recombinases is to resolve chromosome dimers; such dimers would be lethal if unresolved and are formed during replication of circular chromosomes by homologous recombination between the old and the newly synthesized replichore (30, 31). Chromosome dimer resolution by XerC/XerD occurs at the last stage of cell division. This spatial and temporal coordination is accomplished by the divisome protein FtsK (32–34). Its N-terminal domain FtsK$_N$ anchors the protein in the membrane by several membrane-spanning regions. The C-terminal domain of FtsK$_C$ is a DNA translocase which moves chromosomal DNA along small oriented repeats (KOPS) toward *dif* (32, 35, 36). The very end of the C-terminal domain, called FtsK$\gamma$, is required for activation of the XerC/XerD recombinases (32). They bind *dif* to form a pseudotetrameric synaptic complex, but it is only after interaction with FtsK that this complex is catalytically active (31, 32). Dimer resolution by XerC/XerD-*dif*-FtsK results in two intact circular chromosomes, each carrying a *dif* site (30, 31, 33). Although linear chromosomes are found in some bacteria, e.g., *Borrelia* sp., *Streptomyces* sp., *Agrobacterium tumefaciens*, and others (37), circular chromosomes are the rule; therefore, this mechanism is universally required. The *dif* sequence was identified in 641 organisms from 16 phyla, including *D. shibae* (38).

How then could the *dif* sequence and its immediate genetic surroundings be excised and exported into OMVs? While the initiation of replication of circular bacterial chromosomes is well understood (39), many questions remain open regarding the precise mechanisms of its termination (40). Replication starts at *ori* with two replication forks going around the chromosome in opposite directions, i.e., clockwise and counterclockwise (39). Synthesis of the daughter chromosomes exclusively occurs in 5′ to 3′ direction. The leading strand of both forks is synthesized continually, while the lagging strand is synthesized discontinuously through Okazaki fragments (41). Both processes occur simultaneously in the multiprotein complex termed replisome.

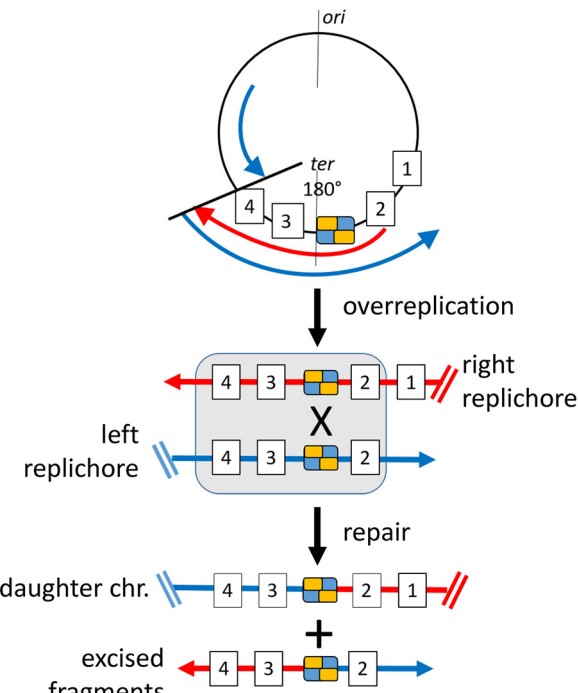

**FIG 3** Working hypothesis for enrichment of *dif*-containing DNA in OMVs of *D. shibae*. The chromosome is schematically shown by a black circle. Origin of replication (*ori*) and terminus region (*ter*) at 180° are indicated. The palindromic binding sequence *dif* is represented by a blue and yellow box, and genes in the terminus region are shown as boxed numbers 1 to 4. Left and right replichores are represented by blue and red lines, respectively. Overreplicated genes are boxed. Arrows represent the replication fork. Only the leading strand of each replichore is shown. chr, chromosome. Two possible mechanisms of repair of overreplicated regions are shown in detail in Fig. S4.

But what exactly happens when the left and right replisomes meet in the terminus region? It is clear that this region is located at 180°, opposite *ori* and characterized by a switch in the GC skew, i.e., the proportion of G and C in the DNA sequence (38). Ideally, the right and left halves of the new chromosomes meet exactly at the terminus and simply ligate (see Fig. S4A). However, the left and right replisome may not reach the 180° point at the same time. DNA synthesis can be delayed because of replication-transcription collision, which has been shown for highly transcribed genes such as ribosomal operons (42). In this case, the replication forks meet before or beyond 180°. Overreplication can be caused by template switching (40, 43, 44) or transient reversal of the replication fork (42).

*Escherichia coli* has a mechanism to prevent overreplication, namely, the so-called "replication fork trap" encoded by Tus (43). However, Tus is not universally present in bacteria but is a recent acquisition in some *Enterobacteriaceae* (45). A deletion of Tus has no phenotype in *E. coli* (43). Also, in *E. coli*, it has been shown that replication forks transiently continue beyond 180° and that the overreplicated regions are excised by an enzymatic system that includes RecBCD but not RecA (40).

We hypothesize that the repair of overreplication accounts for our finding of *dif*-containing DNA fragments in OMVs of *D. shibae* (Fig. 3). Regarding the molecular mechanism of repair, at least two scenarios can be envisaged (Fig. S4).

Scenario 1 (Fig. S4B) postulates that both replication forks meet at *ter* but ligation cannot occur because of template switching, i.e., the leading strand of one fork uses the newly replicated leading strand of the other fork as a template instead of the original chromosome. The overreplicated fragment is excised by an endonuclease that recognizes double-strand breaks, resulting in a linear DNA fragment containing *dif*. Ligation of the double-strand breaks then yields two intact daughter chromosomes.

Scenario 2 (Fig. S4C) postulates that the two replication forks collide outside *ter*, e.g., after overreplication of genes 3 and 4 by the right replisome. Overreplication by template switching and replication fork reversal yields four halves of the daughter chromosomes, each carrying excess genes from the terminus region. This is repaired by site-specific recombination of left and right halves at *dif* via XerC/XerD, resulting in two complete daughter chromosomes plus two copies of excised DNA fragments containing *dif*. This scenario also requires strand breaks and ligation reactions and results in a complex mosaic of overreplicated fragments in the terminus region. According to Occam's razor, therefore, this model might be less likely. It does have the beauty, however, of providing a defined location on the chromosome for repair of overreplication and a precise mechanism for it. Moreover, it accounts for the pronounced peak of vesicle DNA at *dif*.

**The protein inventory of *D. shibae* vesicles is dominated by outer membrane and periplasmic proteins.** Vesicles of *D. shibae* were concentrated and purified from 6 liters of culture in three independent experiments, and the membrane fraction was separated from the soluble protein fraction. Bacterial cells from the same culture were harvested, washed, and subsequently also separated into membrane and soluble fractions. The proteome was determined by gel electrophoresis combined with liquid chromatography-tandem mass spectrometry (GeLC-MS/MS) analyses. Protein quantification was performed using MaxQuant (version 1.5.2.8) intensity-based absolute quantification (iBAQ) (46, 47). Figure 4A shows the overall compositions of proteins in vesicles and whole cells of *D. shibae*, both in the membrane and in the cellular compartment. In *D. shibae* vesicles, 1,393 proteins were identified in the membrane fraction and 2,223 in the soluble fraction (Data Set S2, sheets 1 and 2). In *D. shibae* cells, 1,962 proteins were identified in the membrane fraction and 2,548 in the soluble fraction (Data Set S2, sheets 3 and 4). From the 2,223 proteins identified in the soluble fractions of vesicles, 331 were predicted to be located in the periplasm, covering 32% of the total relative iBAQ value (riBAQ) (48) of this fraction (Fig. 4A). Although the total numbers of periplasmic proteins within the soluble fractions of cells were similar ($n = 302$), these proteins comprised only 10% of the total riBAQ. Similarly, from the 1,393 proteins identified in the membrane fraction of vesicles, 80 were predicted to be OM proteins, representing 48% of the total riBAQ, while the 76 OM proteins in the cell membrane proteome covered only 10% of the total riBAQ. The intensity distribution of OM proteins and extracellular proteins in OMVs was extended by an order of magnitude into values from $10^{-2}$ to $10^{-1}$ riBAQ for a significant amount of proteins compared to that from cells (Fig. 4A). These results clearly show that proteins associated with the outer membrane and the periplasm were strongly enriched in vesicles and confirm that the majority of vesicles were derived from the outer membrane and enclose periplasmic proteins of *D. shibae* cells.

A random forest algorithm was trained to predict the localization of each protein based on riBAQ values of the four samples. The purpose of training the random forest was not the predictions themselves but to identify those proteins that had the highest influence on the random forest in terms of the so-called local importance values (49), which can be derived from the training of the random forest.

Then, proteins of each sample were sorted by their local importance (Data Set S2, sheet 5). Considering the predicted localization of the top 30 proteins (LocateP v2) in the sample "membrane fraction of vesicles," 26 were predicted to be outer membrane proteins, and in the sample "soluble fraction of vesicles," all of these were periplasmatic and outer membrane proteins. At the same time, in the "membrane fraction of cells," inner membrane proteins and, in the "soluble fraction of cells," cytoplasmatic proteins were dominant among the top 30 proteins. These data further confirm our hypothesis that in *D. shibae*, vesicles are derived from the outer membrane and enclose periplasmic proteins.

An enrichment analysis using the 700 proteins with highest riBAQs of each sample revealed that proteins predicted to be localized in the outer membrane or periplasm or to be secreted are more frequently detected in the soluble fraction and membrane fraction of vesicle samples (Data Set S2, sheet 6). An enrichment analysis for functional

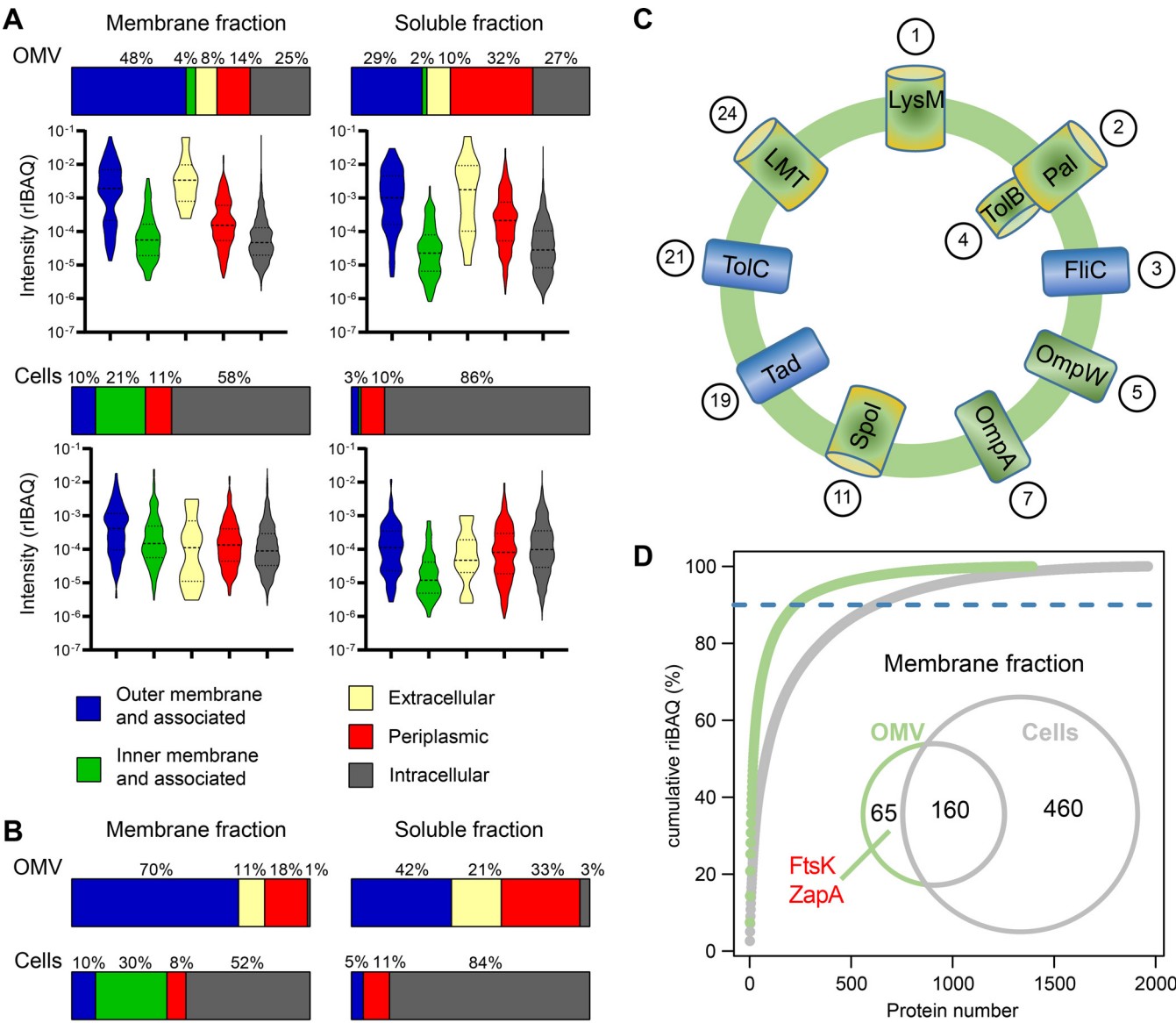

**FIG 4** Vesicles of *D. shibae* are enriched for outer membrane and periplasmic proteins, of which many are related to cell division. Predicted localization of all detected (A) and the 30 most abundant (B) soluble and membrane proteins from *D. shibae* vesicles and cells. (A) Violin plots show the frequency distribution of relative iBAQ values within the groups of each fraction. Median and quantile borders are indicated by dashed and dotted lines, respectively. In total, 1,393 proteins were identified in the membrane fraction of vesicles, 2,223 in the soluble fraction of vesicles, 1,962 proteins in the membrane fraction of cells, and 2,548 proteins in the soluble fraction of cells. (C) Scheme of the most abundant proteins in the OMV membrane. Proteins are ordered in clockwise fashion according to their relative abundance (their rank among the top 30 most abundant vesicle membrane proteins is indicated in circles). All of the predicted functions are hypothetical. LysM, a protein containing a LysM domain required for peptidoglycan hydrolysis; Pal, an outer membrane lipoprotein preferentially located at the septum; TolB, the periplasmic component of the Pal-Tol complex required for cell division; FliC, flagellum filament protein; OmpW, an abundant pore in the outer membrane; OmpA, one of the most abundant outer membrane proteins with two domains, a pore, and a cytoplasmic domain that interacts with peptidoglycan; Spo1, a protein containing a SPOR domain which preferentially binds denuded peptidoglycan at the septum; Tad, part of the Tad (tight adherence) pilus assembly and secretion system common in roseobacters; TolC, the outer membrane component of an energy driven multidrug efflux pump in Gram-negative bacteria; LMT, a lytic murein transglycosylase that divides the septal murein into separate layers. See Results and Discussion for details on those proteins. (D) Abundant OMV membrane proteins. Venn diagram shows unique and overlapping number of proteins comprising 90% of cumulative riBAQ of the respective protein fraction in cells and OMVs. ZapA and FtsK within the OMV fraction are indicated.

categories (again based on the 700 most abundant proteins of each sample) showed that proteins related to envelope biogenesis, integrity, and membranes are prevalent in the soluble and membrane fractions of vesicles (Data Set S2, sheet 7).

We then focused on the 30 most abundant proteins from each fraction, which were identified based on their relative riBAQ value and are shown in Tables 1 to 4 together-with their predicted function, predicted subcellular localization, and relative abundance. The top 30 proteins from the vesicle membrane constituted 59.5% of the total riBAQ of

**TABLE 1** Top 30 most abundant proteins in the vesicle membrane fraction of *D. shibae*

| Locus tag | Function[a] | Subcellular localization[b] | Functional category | riBAQ[c] | | |
|---|---|---|---|---|---|---|
| | | | | Mean[d] | SD | % |
| Dshi_2138 | LysM peptidoglycan-binding domain-containing protein | Periplasmic protein | Not assigned | 7.50E−02 | 1.29E−02 | 7.50 |
| Dshi_1112 | Peptidoglycan-associated lipoprotein | OM lipoprotein | Cell wall/membrane/envelope biogenesis/integrity | 6.82E−02 | 4.78E−03 | 6.82 |
| Dshi_3361 | Flagellin | Secreted/released | Cell motility | 6.49E−02 | 2.27E−02 | 6.49 |
| Dshi_1111 | Tol-Pal system beta propeller repeat TolB | OM-associated/OM $\beta$-barrel protein | Cell wall/membrane/envelope biogenesis/integrity | 4.43E−02 | 9.50E−03 | 4.43 |
| Dshi_1299 | OmpW family protein | OM $\beta$-barrel protein | Not assigned | 2.92E−02 | 9.69E−03 | 2.92 |
| Dshi_1564 | Hypothetical protein | OM lipoprotein | Hypothetical protein | 2.62E−02 | 7.76E−03 | 2.62 |
| Dshi_3025 | OmpA/MotB domain protein | OM lipoprotein | Not assigned | 2.50E−02 | 6.36E−03 | 2.50 |
| Dshi_1128 | OmpA/MotB domain protein | OM lipoprotein | Not assigned | 2.44E−02 | 5.16E−03 | 2.44 |
| Dshi_3376 | Flagellar P-ring protein FlgI | Periplasmic protein | Cell motility | 1.87E−02 | 8.33E−03 | 1.87 |
| Dshi_2832 | Hypothetical protein | OM lipoprotein | Hypothetical protein | 1.68E−02 | 5.68E−03 | 1.68 |
| Dshi_0924 | Sporulation domain protein | OM lipoprotein | Not assigned | 1.68E−02 | 7.46E−03 | 1.68 |
| Dshi_1068 | Hypothetical protein | OM lipoprotein | Hypothetical protein | 1.63E−02 | 5.39E−03 | 1.63 |
| Dshi_2780 | Hypothetical protein | OM lipoprotein | Hypothetical protein | 1.57E−02 | 4.71E−03 | 1.57 |
| Dshi_3379 | Flagellar hook protein FlgE | Cell surface appendage | Cell motility | 1.43E−02 | 1.01E−02 | 1.44 |
| Dshi_1165 | Pyrroloquinoline quinone | OM lipoprotein | Not assigned | 1.32E−02 | 2.42E−03 | 1.32 |
| Dshi_0044 | Hypothetical protein | OM lipoprotein | Hypothetical protein | 1.24E−02 | 1.51E−03 | 1.24 |
| Dshi_1195 | TRAP transporter solute receptor | Periplasmic protein | Transporter | 1.15E−02 | 2.04E−03 | 1.15 |
| Dshi_0056 | Conserved hypothetical protein | OM lipoprotein | Hypothetical protein | 1.10E−02 | 2.24E−03 | 1.10 |
| Dshi_1129 | Type II and III secretion system protein | OM-associated/OM $\beta$-barrel protein | Secretion system | 1.01E−02 | 5.98E−03 | 1.01 |
| Dshi_1351 | Hypothetical protein | OM lipoprotein | Hypothetical protein | 9.30E−03 | 3.04E−03 | 0.93 |
| Dshi_2098 | Putative outer membrane protein TolC | OM-associated/OM $\beta$-barrel protein | Not assigned | 8.36E−03 | 2.04E−03 | 0.84 |
| Dshi_0570 | ABC-type cobalamin/$Fe^{3+}$-siderophores transport | OM-associated/OM $\beta$-barrel protein | Iron metabolism | 8.29E−03 | 6.96E−03 | 0.83 |
| Dshi_1122 | Tetratricopeptide region | OM lipoprotein | Not assigned | 7.82E−03 | 1.20E−03 | 0.78 |
| Dshi_1864 | Putative membrane-bound lytic murein transglycosylase | OM lipoprotein | Not assigned | 7.21E−03 | 1.02E−03 | 0.72 |
| Dshi_2254 | Putative polysaccharide export protein | OM lipoprotein | Transporter | 7.06E−03 | 2.85E−03 | 0.71 |
| Dshi_1499 | Hypothetical protein | OM-associated/OM $\beta$-barrel protein | Hypothetical protein | 6.82E−03 | 2.55E−03 | 0.68 |
| Dshi_3254 | Flagellar basal body L-ring protein | OM lipoprotein | Cell motility | 6.78E−03 | 1.70E−03 | 0.68 |
| Dshi_3495 | Putative SCP-like extracellular protein | OM lipoprotein | Not assigned | 6.59E−03 | 6.86E−04 | 0.66 |
| Dshi_2934 | ATP synthase subunit beta | Intracellular protein | Electron transport | 6.28E−03 | 1.58E−03 | 0.63 |
| Dshi_1483 | Conserved hypothetical protein | OM lipoprotein | Hypothetical protein | 6.15E−03 | 1.29E−03 | 0.62 |

[a]Description of the gene products based on the *D. shibae* DFL 12, DSM 16493 genome annotation (NCBI as of 9 May 2018).
[b]Subcellular localization of the identified proteins predicted using LocateP v2. OM, outer membrane.
[c]riBAQ, relative intensity-based absolute quantification, a normalized measure of molar abundance of a protein calculated by dividing the protein's iBAQ value by the sum of all noncontaminant iBAQ values of a given sample.
[d]The means from three biological replicates are presented.

the respective protein fraction (Fig. 4B). They were clearly dominated by outer membrane proteins ($n = 25$). The top 30 proteins of the soluble fraction were dominated by periplasmic ($n = 10$) and outer membrane ($n = 14$) proteins. In contrast, in the whole-cell proteome, these proteins were rather underrepresented among the top 30 proteins. Here, as expected, we identified mainly intracellular proteins ($n = 26$) within the soluble fraction and intracellular ($n = 16$) and inner membrane ($n = 8$) proteins within the membrane fraction.

There was very little overlap between the top 30 proteins of vesicles and cells. From the top 30 membrane proteins, only two were shared: a hypothetical protein predicted to be an OM lipoprotein (Dshi_1564), and ATP synthase subunit beta (Dshi_2934), an abundant intracellular protein. From the top 30 soluble proteins, an overlap of four proteins was found between vesicles and cells, namely, three solute binding

**TABLE 2** Top 30 most abundant proteins in the vesicle soluble fraction of *D. shibae*

| Locus tag | Function[a] | Subcellular localization[b] | Functional category | riBAQ[c] Mean[d] | SD | % |
|---|---|---|---|---|---|---|
| Dshi_3361 | Flagellin | Secreted/released | Cell motility | 6.85E−02 | 1.23E−02 | 6.85 |
| Dshi_2138 | Hypothetical protein | Periplasmic protein | Hypothetical protein | 3.50E−02 | 1.60E−03 | 3.50 |
| Dshi_3025 | OmpA/MotB domain protein | OM lipoprotein | Not assigned | 3.03E−02 | 6.74E−03 | 3.03 |
| Dshi_1195 | TRAP transporter solute receptor | Periplasmic protein | Transporter | 3.03E−02 | 9.96E−03 | 3.03 |
| Dshi_2021 | Putative binding protein component of ABC iron transporter | OM-associated/OM $\beta$-barrel protein | Iron metabolism | 2.51E−02 | 6.16E−03 | 2.51 |
| Dshi_1111 | Tol-Pal system beta propeller repeat TolB | OM-associated/OM $\beta$-barrel protein | Lipid metabolism and cell wall/membrane/ envelope biogenesis/ integrity | 2.24E−02 | 3.07E−03 | 2.24 |
| Dshi_0318 | Glutamate/glutamine/aspartate/ asparagine ABC transporter | Periplasmic protein | Transporter | 2.20E−02 | 5.41E−03 | 2.20 |
| Dshi_2832 | Hypothetical protein | OM lipoprotein | Hypothetical protein | 1.53E−02 | 9.27E−04 | 1.53 |
| Dshi_2784 | Hypothetical protein | OM lipoprotein | Hypothetical protein | 1.44E−02 | 4.66E−03 | 1.44 |
| Dshi_0924 | Sporulation domain protein | OM lipoprotein | Not assigned | 1.29E−02 | 1.46E−03 | 1.29 |
| Dshi_0563 | Iron-regulated protein | Periplasmic protein | Iron metabolism | 1.20E−02 | 5.76E−03 | 1.20 |
| Dshi_1112 | Peptidoglycan-associated lipoprotein | OM lipoprotein | Lipid metabolism and cell wall/membrane/ envelope biogenesis/ integrity | 1.19E−02 | 8.53E−04 | 1.19 |
| Dshi_1443 | TRAP transporter solute receptor | Periplasmic protein | Transporter | 1.10E−02 | 2.72E−03 | 1.10 |
| Dshi_1299 | OmpW family protein | OM $\beta$-barrel protein | Not assigned | 1.07E−02 | 3.87E−03 | 1.07 |
| Dshi_3872 | Hemolysin-type calcium-binding repeat protein | Secreted/released | Not assigned | 9.83E−03 | 5.91E−03 | 0.98 |
| Dshi_2673 | Quinoprotein ethanol dehydrogenase precursor | Periplasmic protein | Carbon and energy metabolism | 9.59E−03 | 1.14E−03 | 0.96 |
| Dshi_3254 | Flagellar basal body L-ring protein | OM lipoprotein | Cell motility | 9.45E−03 | 2.06E−03 | 0.94 |
| Dshi_1128 | OmpA/MotB domain protein | OM lipoprotein | Not assigned | 9.43E−03 | 2.22E−03 | 0.94 |
| Dshi_1522 | Putative periplasmic ligand-binding protein | Periplasmic protein | Transporter | 9.42E−03 | 3.07E−03 | 0.94 |
| Dshi_1622 | Putative hemolysin precursor | Secreted/released | Not assigned | 9.37E−03 | 3.33E−03 | 0.94 |
| Dshi_1483 | Conserved hypothetical protein | OM lipoprotein | Hypothetical protein | 9.19E−03 | 1.59E−03 | 0.92 |
| Dshi_3402 | Neutral zinc metallopeptidase | Secreted/released | Protein fate | 8.99E−03 | 3.61E−03 | 0.90 |
| Dshi_2763 | Hypothetical protein | Intracellular protein | Hypothetical protein | 8.94E−03 | 3.35E−03 | 0.89 |
| Dshi_2343 | Hypothetical protein | OM-associated/OM $\beta$-barrel protein | Hypothetical protein | 8.93E−03 | 1.82E−03 | 0.89 |
| Dshi_1564 | Hypothetical protein | OM lipoprotein | Hypothetical protein | 7.91E−03 | 2.69E−03 | 0.79 |
| Dshi_3153 | TRAP transporter, DctP subunit | Periplasmic protein | Transporter | 7.85E−03 | 1.37E−03 | 0.79 |
| Dshi_0872 | Extracellular solute-binding protein family 5 | Periplasmic protein | Transporter | 7.79E−03 | 2.35E−03 | 0.78 |
| Dshi_0274/Dshi_0223 | Translation elongation factor Tu | Intracellular protein | Translation | 7.14E−03 | 2.36E−03 | 0.71 |
| Dshi_0628 | Basic organic compound ABC-transporter | Periplasmic protein | Transporter | 6.94E−03 | 2.72E−03 | 0.69 |
| Dshi_2780 | Hypothetical protein | OM lipoprotein | Hypothetical protein | 6.81E−03 | 1.59E−03 | 0.68 |

[a]Description of the gene products based on the *D. shibae* DFL 12, DSM 16493 genome annotation (NCBI as of 9 May 2018).
[b]Subcellular localization of the identified proteins predicted using LocateP v2. OM, outer membrane.
[c]riBAQ, relative intensity-based absolute quantification, a normalized measure of molar abundance of a protein calculated by dividing the protein's iBAQ value by the sum of all noncontaminant iBAQ values of a given sample.
[d]The means from three biological replicates are presented.

components of ABC transporters (Dshi_1195, Dshi_2021, and Dshi_0318) and the translation elongation factor Tu.

**The most abundant vesicle membrane proteins are predicted to be involved in cell division.** The top 30 vesicle membrane proteins (Table 1) will now be discussed in the order of their abundance and are schematically shown in Fig. 4C. With the exception of LysM, all of them were also identified by the random forest analysis of predictive capability of proteins for localization in the vesicle membrane (Data Set S2, sheet 5). The most abundant vesicle membrane protein (Dshi_2138), comprising 7.5% of the total riBAQ, was a hypothetical periplasmic protein predicted to contain a LysM peptidoglycan-binding domain. LysM domains are capable of noncovalent binding to

**TABLE 3** Top 30 most abundant proteins in the cell membrane fraction of *D. shibae*

| Locus tag | Function[a] | Subcellular localization[b] | Functional category | riBAQ[c] Mean[d] | SD | % |
|---|---|---|---|---|---|---|
| Dshi_3540 | Reaction center protein H chain | Intracellular protein | Electron transport | 2.63E−02 | 1.30E−03 | 2.63 |
| Dshi_2735 | Protein HflC | IM protein (possibly C-terminally anchored) | Protein fate | 2.43E−02 | 6.14E−03 | 2.43 |
| Dshi_1233 | NAD(P) transhydrogenase, alpha subunit | Integral IM protein | Electron transport | 2.12E−02 | 1.14E−03 | 2.12 |
| Dshi_2021 | Putative binding protein component of ABC iron | OM-associated/OM β-barrel protein | Iron metabolism | 1.85E−02 | 1.32E−03 | 1.85 |
| Dshi_3027 | ATP synthase F0 | Integral IM protein | Electron transport | 1.78E−02 | 1.22E−03 | 1.78 |
| Dshi_0664 | Cytochrome *c* oxidase | Intracellular protein | Electron transport | 1.62E−02 | 2.77E−03 | 1.62 |
| Dshi_3525 | Photosynthetic reaction center cytochrome c | Periplasmic protein | Electron transport | 1.52E−02 | 3.76E−03 | 1.52 |
| Dshi_2934 | ATP synthase subunit beta | Intracellular protein | Electron transport | 1.32E−02 | 8.65E−04 | 1.32 |
| Dshi_1860 | Preprotein translocase subunit | Intracellular protein | Secretion system | 1.26E−02 | 7.55E−03 | 1.26 |
| Dshi_2935 | ATP synthase gamma chain | Intracellular protein | Electron transport | 1.10E−02 | 6.96E−05 | 1.10 |
| Dshi_3522 | Light-harvesting protein B-870 alpha chain | Intracellular protein | Electron transport | 1.08E−02 | 9.93E−03 | 1.08 |
| Dshi_3028 | ATP synthase F0 | Intracellular protein | Electron transport | 1.08E−02 | 4.64E−03 | 1.08 |
| Dshi_3278 | Ubiquinol-cytochrome c reductase | Periplasmic protein | Electron transport | 1.06E−02 | 4.51E−03 | 1.06 |
| Dshi_3385 | Tetratricopeptide | OM lipoprotein | Not assigned | 9.80E−03 | 1.16E−03 | 0.98 |
| Dshi_3523 | Reaction center protein L chain | Integral IM protein | Electron transport | 9.77E−03 | 5.80E−03 | 0.98 |
| Dshi_2736 | Protein HflK | Intracellular protein | Protein fate | 9.32E−03 | 1.09E−03 | 0.93 |
| Dshi_0328 | Preprotein translocase subunit | Integral IM protein | Secretion system | 9.21E−03 | 9.33E−04 | 0.92 |
| Dshi_2936 | ATP synthase subunit alpha | Intracellular protein | Electron transport | 8.66E−03 | 2.13E−03 | 0.87 |
| Dshi_2931 | Ribose-phosphate pyrophosphokinase | Intracellular protein | Carbon and energy metabolism | 7.77E−03 | 9.62E−04 | 0.78 |
| Dshi_0295 | 50S ribosomal protein L14 | Intracellular protein | Translation | 7.51E−03 | 6.16E−04 | 0.75 |
| Dshi_0289 | 50S ribosomal protein L16 | Intracellular protein | Translation | 7.39E−03 | 1.75E−03 | 0.74 |
| Dshi_1314 | Conserved hypothetical protein | Integral IM protein | Hypothetical protein | 6.88E−03 | 1.27E−03 | 0.69 |
| Dshi_3590 | Ubiquinone dependent NADH dehydrogenase | Intracellular protein | Electron transport | 6.87E−03 | 1.88E−03 | 0.69 |
| Dshi_3556 | Sodium/solute symporter family protein | Integral IM protein | Transporter | 6.83E−03 | 2.07E−03 | 0.68 |
| Dshi_1564 | Hypothetical protein | OM lipoprotein | Hypothetical protein | 6.72E−03 | 2.89E−03 | 0.67 |
| Dshi_2919 | 60-kDa chaperonin | Intracellular protein | Protein fate | 6.71E−03 | 2.14E−03 | 0.67 |
| Dshi_1234 | NAD(P)$^+$ transhydrogenase | Integral IM protein | Carbon and energy metabolism | 5.99E−03 | 2.34E−03 | 0.60 |
| Dshi_2933 | ATP synthase epsilon chain | Intracellular protein | Electron transport | 5.84E−03 | 1.39E−03 | 0.58 |
| Dshi_3417 | Hypothetical protein | Intracellular protein | Hypothetical protein | 5.79E−03 | 8.12E−04 | 0.58 |
| Dshi_1643 | Bacterial DNA recombination | Intracellular protein | Replication, recombination and repair | 5.68E−03 | 7.18E−04 | 0.57 |

[a]Description of the gene products based on the *D. shibae* DFL 12, DSM 16493 genome annotation (NCBI as of 9 May 2018).
[b]Subcellular localization of the identified proteins predicted using LocateP v2. OM, outer membrane.
[c]riBAQ, relative intensity-based absolute quantification, a normalized measure of molar abundance of a protein calculated by dividing the protein's iBAQ value by the sum of all noncontaminant iBAQ values of a given sample.
[d]The means from three biological replicates are presented.

peptidoglycan by interacting with *N*-acetylglucosamine moieties; the motif is found in enzymes from all domains of life (Pfam database PF01476) (50). Many of the proteins containing LysM domains are cell wall hydrolases that require LysM for proper positioning of the active site toward their substrate (50). Thus, the LysM-containing hypothetical protein might be involved in cell wall remodeling during growth.

The Pal (Dshi_1112) (6.8%) and TolB (Dshi_1111) (4.4%) proteins belong to the Tol-Pal complex that spans the cell envelopes of Gram-negative bacteria and coordinates outer membrane constriction with septation during cell division (51–54). The Tol-Pal complex consists of the three inner membrane (IM) proteins TolA, TolR, and TolQ, the periplasmic protein TolB, and the outer membrane lipoprotein Pal (55). The IM proteins TolA, TolR, and TolQ were not among the top 30 proteins of the vesicle membranes,

**TABLE 4** Top 30 most abundant proteins in the cell soluble fraction of *D. shibae*

| Locus tag | Function[a] | Subcellular localization[b] | Functional category (IMG) | riBAQ[c] Mean[d] | SD | % |
|---|---|---|---|---|---|---|
| Dshi_2919 | 60-kDa chaperonin | Intracellular protein | Protein fate | 2.24E−02 | 1.68E−03 | 2.24 |
| Dshi_0274/Dshi_0223 | Translation elongation factor Tu | Intracellular protein | Translation | 2.04E−02 | 6.66E−03 | 2.04 |
| Dshi_2934 | ATP synthase subunit beta | Intracellular protein | Electron transport | 1.41E−02 | 1.90E−03 | 1.41 |
| Dshi_2021 | Putative binding protein component of ABC iron transporter | OM-associated/OM β-barrel protein | Iron metabolism | 1.22E−02 | 1.56E−03 | 1.22 |
| Dshi_1350 | Nucleoside-diphosphate kinase | Intracellular protein | Not assigned | 1.11E−02 | 1.43E−03 | 1.11 |
| Dshi_1195 | TRAP transporter solute receptor | Periplasmic protein | Transporter | 9.51E−03 | 1.98E−03 | 0.95 |
| Dshi_0318 | Glutamate/glutamine/aspartate/ asparagine ABC transporter | Periplasmic protein | Transporter | 9.23E−03 | 1.33E−03 | 0.92 |
| Dshi_0762 | Aminotransferase class I and II | Intracellular protein | Amino acid metabolism | 7.45E−03 | 1.21E−03 | 0.75 |
| Dshi_0273 | Translation elongation factor G | Intracellular protein | Translation | 7.22E−03 | 1.46E−03 | 0.72 |
| Dshi_3318 | D-3-Phosphoglycerate dehydrogenase | Intracellular protein | Amino acid metabolism | 6.47E−03 | 5.58E−04 | 0.65 |
| Dshi_3067 | Acetoacetyl coenzyme A reductase | Intracellular protein | Carbon and energy metabolism | 6.34E−03 | 7.40E−04 | 0.63 |
| Dshi_2190 | Trigger factor | Intracellular protein | Not assigned | 6.30E−03 | 1.18E−03 | 0.63 |
| Dshi_1549 | Translation elongation factor | Intracellular protein | Translation | 6.30E−03 | 9.74E−04 | 0.63 |
| Dshi_1012 | Methionine adenosyltransferase | Intracellular protein | Amino acid metabolism | 6.05E−03 | 6.51E−04 | 0.61 |
| Dshi_0216 | Electron transfer flavoprotein alpha subunit | Intracellular protein | Electron transport | 5.90E−03 | 4.98E−04 | 0.59 |
| Dshi_1839 | Glutamine synthetase | Intracellular protein | Amino acid metabolism | 5.74E−03 | 5.75E−04 | 0.57 |
| Dshi_2936 | ATP synthase subunit alpha | Intracellular protein | Electron transport | 5.55E−03 | 5.35E−04 | 0.56 |
| Dshi_2705 | Uroporphyrinogen decarboxylase | Intracellular protein | Porphyrin and related pathway | 5.51E−03 | 8.56E−04 | 0.55 |
| Dshi_0006 | 2,3,4,5-Tetrahydropyridine-2,6-dicarboxylate N-succinyltransferase | Intracellular protein | Amino acid metabolism | 5.34E−03 | 5.62E−04 | 0.53 |
| Dshi_0950 | Ribosomal 5S rRNA E-loop binding protein | Intracellular protein | Translation | 5.30E−03 | 1.12E−03 | 0.53 |
| Dshi_3571 | Chaperone protein DnaK | Intracellular protein | Protein fate | 5.13E−03 | 8.66E−04 | 0.51 |
| Dshi_0081 | 3-Isopropylmalate dehydrogenase | Intracellular protein | Amino acid metabolism | 5.12E−03 | 1.58E−03 | 0.51 |
| Dshi_2704 | Porphobilinogen deaminase | Intracellular protein | Porphyrin and related pathway | 4.96E−03 | 1.90E−03 | 0.50 |
| Dshi_3165 | Periplasmic nitrate reductase NapA | Periplasmic protein | Electron transport | 4.83E−03 | 1.30E−03 | 0.48 |
| Dshi_0217 | Electron transfer flavoprotein beta subunit | Intracellular protein | Electron transport | 4.66E−03 | 6.15E−04 | 0.47 |
| Dshi_3066 | Acetyl coenzyme A acetyltransferase | Intracellular protein | Lipid metabolism and cell wall/membrane/ envelope biogenesis/ integrity | 4.41E−03 | 9.78E−04 | 0.44 |
| Dshi_0821 | Glycine hydroxymethyltransferase | Intracellular protein | Amino acid metabolism | 4.26E−03 | 8.83E−04 | 0.43 |
| Dshi_3426 | Adenosylhomocysteinase | Intracellular protein | Amino acid metabolism | 4.25E−03 | 3.04E−04 | 0.43 |
| Dshi_2156 | Fructose-bisphosphate aldolase | Intracellular protein | Carbon and energy metabolism | 4.18E−03 | 6.10E−04 | 0.42 |
| Dshi_0825 | AMP-dependent synthetase and ligase | Intracellular protein | Not assigned | 4.16E−03 | 9.67E−04 | 0.42 |

[a]Description of the gene products based on the *D. shibae* DFL 12, DSM 16493 genome annotation (NCBI as of 9 May 2018).
[b]Subcellular localization of the identified proteins predicted using LocateP v2. OM, outer membrane.
[c]riBAQ, relative intensity-based absolute quantification, a normalized measure of molar abundance of a protein calculated by dividing the protein's iBAQ value by the sum of all noncontaminant iBAQ values of a given sample.
[d]The means from three biological replicates are presented.

again confirming that the vesicles were derived from the outer membrane. The strong enrichment of Pal and TolB in vesicles is in accordance with our hypothesis that vesicle biogenesis is related to cell division and septation.

The filament protein flagellin (FliC, Dshi_3361) comprised 6.49% of the vesicle membrane proteins and was the most abundant soluble vesicle protein. *D. shibae* has a polar flagellum, and its synthesis during cell division requires export of flagellin from the cytoplasm; thus, it could be captured in OMVs. We also found flagellar P-ring protein FlgI, flagellar hook protein FlgE, and flagellar basal body L-ring protein, which are located in the OM, as well as the basal body MotB, located in the peptidoglycan layer, but none of the components located in the cytoplasmic membrane. Flagellin is also an abundant component of *E. coli* OMVs, and a *fliC* null mutant produces less OMVs (56).

Various porin-forming proteins were among the most abundant proteins in vesicle membranes, namely, OmpW family protein (Dshi_1299), OmpA (Dshi_3025), and OmpA family protein (Dshi_1128). OmpA is among the most abundant protein in *E. coli* (57). The N-terminal domain forms an eight-stranded beta-barrel with a hydrophobic channel for the diffusion of small molecules. The C-terminal domain is located in the periplasm, and it noncovalently binds to peptidoglycan (58). OmpA can also form a complex with Pal (53) and may play a role in modifying the gap between the outer membrane and peptidoglycan (59).

The sporulation domain protein (Dshi_0924) (1.7% and 1.3% in the vesicle membrane and soluble fractions, respectively) contains a widely conserved peptidoglycan-binding domain important for cell division. The SPOR domain specifically binds to regions of the peptidoglycan which are "denuded," i.e., devoid of stem peptides. In contrast to most other septal ring proteins, it interacts directly with the peptidoglycan (PG) rather than with other enzymes of the divisome (60).

The Tad pilus comprised 1% of vesicle membrane proteins. Dshi_1129 was originally annotated as a type II and III secretion system, but a recent reannotation identified it as part of the Tad (tight adherence) pilus assembly and secretion system (29, 61). It is composed of an assembly platform in the IM and a filament that traverses the peptidoglycan cell wall through a gated pore (62).

TolC (Dshi_2098) is the outer membrane component of an energy-driven multidrug efflux pump in Gram-negative bacteria that is also referred to as a type I secretion system (63). While the Tad systems secrete their cargo in a highly regulated fashion and do not permanently open a pore in the OM, type I systems represent stable channels and are an important part of the intrinsic resistance against antibiotics, e.g., in *Pseudomonas aeruginosa* (63).

Finally, the putative membrane bound lytic murein transglycosylase (LMT) (Dshi_1864) (0.7%) divides the septal murein into separate peptidoglycan layers by reducing cross-links and is required for septum formation and OMV biogenesis (4).

To identify proteins that may be less abundant but still overrepresented in the vesicle membrane proteome, we performed an additional analysis. The riBAQ values from the vesicle membrane proteome and the cell membrane proteome were sorted according to relative abundance, and their cumulative abundance was calculated. Figure 4D shows that both membrane proteomes are dominated by a few very abundant proteins. The cell membrane proteome is much more diverse than that of the OMVs. Four hundred sixty proteins are required to obtain 90% cumulative riBAQ in the cell membrane proteome, but only 225 proteins comprise 90% of the vesicle membrane proteome. We compared these two data sets of highly abundant proteins. Of these, 160 proteins are shared with the cell membrane, while 65 proteins are unique for vesicle membranes (Data Set S2, sheet 8). Among these proteins, we found two divisome components: ZapA (Dshi_1736) and FtsK (Dshi_0059), the DNA translocase that is required to activate the site-specific recombinases XerC/XerD. Both of these enzymes are not part of the outer membrane but are recruited to the inner membrane specifically during cell division. Finding them in the vesicle membrane supports our hypothesis that vesicle formation is coupled to cell division in *D. shibae*.

**The most abundant soluble proteins of *D. shibae* vesicles are transporters and potential virulence factors.** The largest functional group among the soluble vesicle proteins were substrate-binding proteins belonging to high-affinity transport systems, e.g., ATP-binding cassette (ABC) transporters (64) and ATP-independent TRAP transporters (65), the latter being especially abundant in *D. shibae* (19). The binding proteins were predicted to be specific for ferric iron (Dshi 2021), $C_4$-dicarboxylates (Dshi_1195, Dshi_1443, and Dshi_3153), amino acids (Dshi_0318 and Dshi_1522), peptides (Dshi_0872), and sulfate (Dshi_0626).

Finally, proteins with similarity to imelysin (Dshi_0563), serralysin (Dshi_3402), and hemolysin (Dshi_3872, Dshi_1622), which might play a role for interactions of *D. shibae* with the algal host, were enriched in the vesicle lumen. Imelysin-like proteins are hypothesized to bind iron or an unknown ligand (66). Serralysin-like proteins belong

**TABLE 5** Fatty acid composition of membranes from *D. shibae* cells and vesicles

| | Amt of fatty acid (arbitrary units) in: | | | | | | |
|---|---|---|---|---|---|---|---|
| | Cells | | | Vesicles | | | |
| Fatty acid | Avg | SD | % | Avg | SD | % | Fold enrichment in vesicles |
| 10:0 3OH | 1,861 | 86 | 1 | 6,677 | 1,058 | 4 | 3.59 |
| 12:1ω7c | 3,115 | 84 | 2 | 10,156 | 847 | 7 | 3.26 |
| 14:0 3OH | 449 | 37 | 0 | ND[a] | ND | ND | ND |
| 16:1ω7c | 887 | 9 | 1 | ND | ND | ND | ND |
| 16:00 | 2,091 | 30 | 1 | 30,864 | 4,627 | 21 | 14.76 |
| 18:1ω7c | 119,520 | 429 | 80 | 73,593 | 7,076 | 49 | 0.62 |
| 18:00 | 7,687 | 276 | 5 | 28,710 | 13.172 | 19 | 3.73 |
| 11 Me 18:1ω7c | 11,657 | 377 | 8 | ND | ND | ND | ND |
| Unknown | 2,195 | 276 | 1 | ND | ND | ND | ND |
| 20:1ω7c | 538 | 23 | 0 | ND | ND | ND | ND |
| Total signal | 150,000 | | | 150,000 | | | |

[a]ND, not detected.

into the zinc-containing subfamily of extracellular metalloproteases and play fundamentally important roles in pathogenicity (67). The *Bacteroides fragilis* toxin (which is a zinc-dependent nonlethal metalloprotease) is delivered via OMVs to epithelial cells rather than being excreted directly into the extracellular medium (68). The vesicle lumen was enriched for two hemolysin-like calcium-binding proteins. In enterohemorrhagic *Escherichia coli* (EHEC), hemolysins are among the virulence factors which are delivered via OMVs to the host (69).

***D. shibae* OMVs are strongly enriched for long-chain saturated fatty acids.** The fatty acid compositions of membranes from whole cells and vesicles of *D. shibae* are shown in Table 5. Despite different cultivation conditions, it was similar to that determined previously (18), with C18:1ω7c as the dominant fatty acid comprising 80% of all detected fatty acids. Strikingly, in OMV membranes, the proportion of C18:1ω7c was reduced to 49%, while the saturated fatty acids hexadecanoic acid ($C_{16:0}$) and octadecanoic acid (C18:0), which together were below 5% in the overall fatty acid profile, increased to 21% and 19%, respectively. This represents enrichments of 14.75-fold and 3.73-fold, respectively.

The outer membrane (OM) of a Gram-negative bacterium is highly asymmetric (70). Its outer leaflet consists of lipopolysaccharide (LPS), while phospholipids comprise the inner leaflet (and the two leaflets of the cytoplasmic membrane). The LPS consists of three parts from the outside to the inside: a polysaccharide chain (the O-antigen), a core oligosaccharide, and lipid A. Lipid A is composed of a saccharide head acylated with a characteristic set of fatty acids (70, 71). The number and substitutions of the fatty acids in lipid A thus have a profound influence on the overall fatty acid profile. The lipid A composition was determined for *Roseobacter denitrificans*, a close relative of *D. shibae*, and the major fatty acids are C10:0 3OH and C14:03oxo (72). C18:1ω7c, comprising 80% of all fatty acids in both *D. shibae* and *R. denitrificans* (18), is therefore unlikely to be a component of lipid A.

An enrichment of saturated fatty acids in OMVs compared to that in the OM was also found in *E. coli* (73) and *P. aeruginosa* (74). A recent study of OMV biogenesis in *Haemophilus influenzae* found an enrichment of C14:0 and C16:0 in mutants defective for the dedicated VacJ/Yrb ABC transport system (75). The authors proposed a potentially universal model for OMV biogenesis, which suggests that phospholipids naturally accumulate in the outer leaflet of the OM and that this triggers vesicle formation unless the phospholipids are transported back to the inner leaflet by a dedicated transporter system (75, 76). This model would be in accordance with an enrichment of C16:0 and C18:0 in the vesicles. The release of vesicles with more rigid saturated fatty acids than the OM may aid the division process by increasing the fluidity of the OM of the cell.

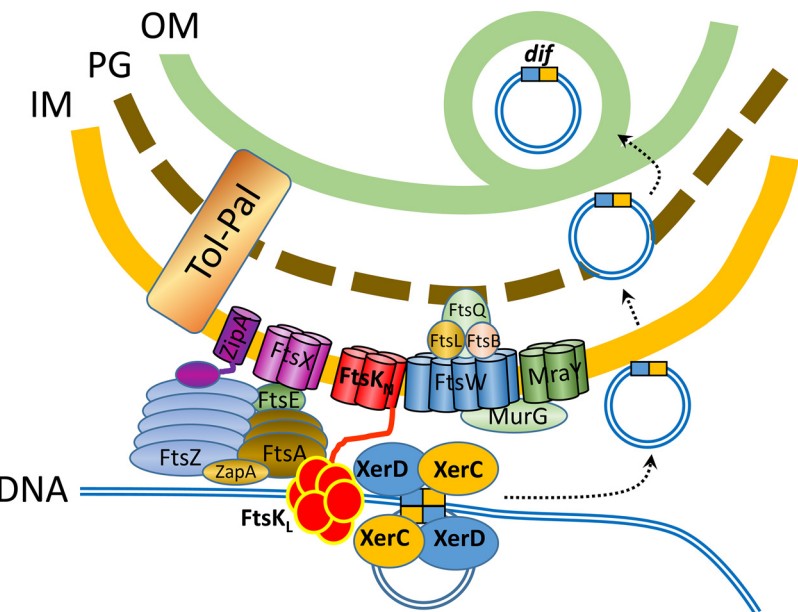

**FIG 5** Scheme of the FtsK-*dif*-XerC/XerD protein complex in the divisome and the export of DNA into OMVs. A subset of the proteins comprising the divisome and their localization at the cell envelope is schematically shown. FtsK$_N$ is the N-terminal domain of FtsK, and FtsK$_L$ is its C-terminal domain. FtsK is a DNA translocase that moves the replichore toward *dif* and activates the site-specific recombinases XerC/XerD. We hypothesize that these enzymes excise overreplicated genes around the terminus which then enter the periplasm and are enclosed by the outer membrane. It is unknown if the excised DNA is circular or linear. OM, outer membrane; PG, peptidoglycan; IM, inner membrane. Modified from references 89 and 90 with incorporation of information from references 34, 35 and 91 to 94.

**Conclusion and outlook.** OMVs are produced by *D. shibae* constitutively during undisturbed growth. They contain DNA which is enriched for the region around the hypothetical terminus of replication, specifically, the 28-bp *dif* site. OMVs are enriched for proteins from the outer membrane and the periplasm. Some of the most abundant membrane proteins of OMVs are predicted to be involved in cell division, one of them interacting directly with denuded peptidoglycan at the septum. Two proteins from the divisome complex (FtsK and ZapA) were enriched in the vesicle proteome. The fatty acid composition of the OMVs differed strikingly from that of the releasing cell, containing a higher percentage of saturated fatty acids.

Our hypothesis for vesicle biogenesis and is schematically shown in Fig. 5. OMVs of *D. shibae* are formed when the replisome and divisome multiprotein nanomachines meet at the division plane. The divisome protein complex spans inner and outer membranes and contains enzymes for septal peptidoglycan synthesis and hydrolysis as well as DNA translocases (34). We suggest that the DNA around *ter* is incorporated into OMVs during the invagination of the membrane that precedes septum formation and that this incorporation is coupled to the activity of the FtsK-XerC/XerD dimer resolution machinery.

Our study leaves two main questions for further analysis. (i) How are the excised fragments transported into the periplasm, where they can be incorporated by the OM? (ii) Which biochemical mechanism accounts for the excision of chromosomal fragments around *dif*?

Regarding the first question: How is DNA that has been excised from the chromosome transported across the IM into the periplasm? Known transport systems span the entire cell envelope and export DNA from the cytoplasm across both the IM and OM into the environment, e.g., TolC, Tad, the flagellar biosynthesis machinery, and type IV secretion systems. TolC, Tad, and FliC were found in the vesicle proteome, together with several porins. Thus, it remains to be investigated if vesicles actually took up DNA

fragments released via the above-mentioned mechanisms from the cytoplasm into the extracellular environment of the cells. An alternative hypothesis is related to our experimental observation of a striking difference between the fatty acid composition of the OMVs and that of the releasing cells. This is in accordance with their release during Z-ring formation, which was reported to result in an altered composition of the membrane phospholipids (77). OMVs contained a higher percentage of saturated fatty acids; thus, they were more rigid than the membranes of the releasing cells, which would support their quick release. We hypothesize that during septum formation, the penetration of the IM was briefly possible for small DNA fragments excised by an enzyme complex localized at the inner membrane close to *dif*, the site of daughter cell separation (78).

For the second question—which mechanism accounts for the excision of chromosomal fragments around *dif*—we have developed two scenarios based on overreplication, of which one relies on endonucleases and ligases, while the other requires an active role of the FtsK-XerC/XerD enzymes. It will be interesting to test those scenarios experimentally and to investigate if additional mechanisms, e.g., homologous recombination, contribute to the observed phenomenon. If an active role of the FtsK-XerC/XerD site-specific recombination could be confirmed, it would be interesting to analyze how conserved this mechanism actually is. Enrichment of the replication terminus has also been observed in OMVs from *Prochlorococcus* (8). In *P. aeruginosa*, enrichment of several genes was reported in OMVs but not of a certain chromosomal region (26). The *dif*-related machinery was identified in 641 organisms from 16 phyla (38). It might have an additional role beyond resolution of chromosome dimers, namely, repair of overreplication, which could be universally required for correct termination of chromosome replication. Transient overreplication might in fact be a mechanism to protect the cell from loss of genetic information (40). These are exciting speculations; current technical developments, including deep sequencing, chromosome conformation capture, and superresolution microscopy, will undoubtedly help to resolve them and provide a much deeper understanding of replication termination in the future.

## MATERIALS AND METHODS

**Strain and cultivation conditions.** All experiments were performed with *D. shibae* DSM 16493[T] cultivated in defined saltwater medium (SWM) (18) using 20 mM succinate as carbon source. Liquid cultures were incubated at 30°C in the dark on a platform shaker with 160 rpm. Each experiment was started by streaking cells from glycerol storage cultures on marine broth (MB) plates. After 3 to 4 days, cell material was transferred to a liquid preculture in 25 ml SWM plus 20 mM succinate and grown for approximately 24 h. A second preculture was prepared with a volume of 100 ml in a 300-ml flask with an initial optical density at 600 nm ($OD_{600}$) of 0.03. The second preculture was used for preparing four main cultures with a 1.5-liter culture volume each in 3-liter flasks at an initial $OD_{600}$ of 0.03. Cultures were harvested at an $OD_{600}$ of around 2.5, which represents the late-exponential growth phase.

**Vesicle concentration and purification.** Bacterial cells were removed by centrifugation at 10,900 × *g* for 15 min at 4°C. The supernatant was filtered through 0.45-$\mu$m (Nalgene, Thermo Scientific) and 0.22-$\mu$m (Millipore) bottle-top filters. The filtrate was concentrated using a tangential flow filtration system (Vivaflow 200; Sartorius) with a 100-kDa-molecular-mass cutoff. The concentrate was ultracentrifuged at 100,000 × *g* for 2 h at 4°C. The pellets were resuspended in 2 ml 45% OptiPrep (in buffer containing 3.6% [wt/vol] NaCl and 10 mM HEPES, pH 8). Samples were loaded into the bottom of a 13.2-ml ultracentrifuge tube and overlaid with 1 ml of 40%, 35%, 30%, 25%, 20%, 15%, 10%, 5%, and 0% OptiPrep. The gradient was centrifuged at 280,000 × *g* for 3 to 12 h at 4°C. The brownish fraction containing pure vesicles (Fig. 1A) was collected, diluted with buffer (10 mM HEPES, 3.6% NaCl) to 30 ml and pelleted at 100,000 × *g* for 2 h at 4°C. The supernatant was discarded, and the final pellets were frozen at −70°C until further analysis.

**Particle-tracking analysis using the NanoSight instrument.** For determining the size spectrum and abundance of vesicles, particle-tracking analysis was performed using the NanoSight NS300 (Malvern Panalytical) equipped with a 488-nm laser, a flow cell, and a syringe pump. Samples were imaged under light-scatter mode with camera level 15 and a detection limit of 5. We measured the particle concentration of the buffer used to resuspend the OMVs after gradient centrifugation. The buffer contained $2.31 \times 10^6 \pm 7.63 \times 10^5$ particles/ml with an average size of 71.7 ± 4.4 nm. The particle count in the buffer was therefore considered negligible compared to the $10^9$ to $10^{10}$ particles/ml in the vesicle preparation. Samples were diluted in 0.22-$\mu$m filtered SWM until approximately 30 vesicles were present in the viewing field of the instrument. Three videos of 1-min duration were taken per sample, and the mean and standard deviation were calculated. Concentrated and purified vesicles used for proteome analysis were imaged after 10,000-fold dilution.

To follow OMV production during growth, the concentration of OMVs in the supernatant of cultures was determined. Three 100-ml cultures of *D. shibae* were inoculated to an $OD_{600}$ of 0.03 and served as three biological replicates. A 1-ml sample was taken from each culture to determine the cell count by flow cytometry as described previously (27), and a second 1-ml sample was filtered through a 0.22-$\mu$m syringe filter (Carl-Roth) and used to determine the vesicle count using the NanoSight instrument by recording 3 videos as described above. Samples were taken over a period of 40 h mainly in 2-h intervals.

**Time-lapse microscopy.** Time-lapse microscopy was performed as described before (79). Briefly, 2 $\mu$l of *D. shibae* cells of an exponentially growing culture were immobilized on a 1% agarose pad containing SWM (20 mM succinate), and cells were imaged in a MatTek glass-bottom microwell dish (35-mm dish, 14-mm microwell with number [no.] 1.5 cover glass, P35G-1.5-14-C). Images were taken with phase-contrast illumination using a NikonTi microscope with a Nikon N Plan Apochromat $\lambda$ 100$\times$/1.45 numerical aperture oil immersion lens objective and an ORCA-Flash 4.0 Hamamatsu camera. Cell growth and vesicle formation were observed every 5 min for 24 h at 30°C. Micrographs were subsequently aligned and analyzed using the NIS-Elements imaging software V 4.3 (Nikon).

**Wide-field fluorescence microscopy.** To detect the putative presence of DNA on and within the OMVs, isolated OMVs were treated with FM1-43 {*N*-(3-triethylammoniumpropyl)-4-[4-(dibutylamino)styryl] pyridinium dibromide} and DAPI (4',6-diamidino-2-phenylindole) at final concentrations of 1 $\mu$M and 3 $\mu$g/ml, respectively. OMVs were incubated for 10 min at room temperature (RT) in the dark and were washed twice with 1 ml 1$\times$ phosphate-buffered saline (PBS) (centrifugation at 14,000 rpm for 3 min). For visualization, OMVs were immobilized on a 1% agarose pad in a MatTek glass-bottom microwell dishes (35-mm dish, 14-mm microwell with no. 1.5 cover glass, P35G-1.5-14-C) as described before (80). Wide-filed (WF) fluorescence micrographs were obtained with DAPI (370/36 nm to 440/40 nm) and green fluorescent protein (GFP) (485/20 nm to 525/30 nm) filters. Fluorescence z-stacks and phase-contrast images were taken using a Nikon N Plan Apochromat $\lambda$ 100$\times$/1.45 numerical aperture oil immersion lens objective and an ORCA-Flash 4.0 Hamamatsu camera. Images were processed using the NIS-elements imaging software V 4.3 (Nikon) together with the three-dimensional (3D) Landweber deconvolution algorithm (z-step, 0.2 $\mu$m; spherical aberration, 0.2). For quantification, DNase (DNase I, RNase free, 1000 U, EN0521; Thermo Scientific)-treated OMVs (0.5 U/ml, 30 min, 37°C) were stained with FM1-43 and DAPI at a final concentration of 3 $\mu$g/ml. OMV fluorescence signals were determined by the object count tool of the NIS-Elements imaging software V 4.3 (Nikon). In total, >20,000 OMV in 20 fields of view in 2 independent staining experiments were quantified.

**Electron microscopy.** Scanning electron microscopy (SEM) was performed as previously described (81). Briefly, samples were placed onto poly-L-lysine-coated coverslips (12 mm) for 10 min, fixed with 2% glutaraldehyde in TE buffer (10 mM Tris, 1 mM EDTA, pH 6.9), and dehydrated with a graded series of acetone (10, 30, 50, 70, 90, and 100%) on ice, 10 min for each step. After critical-point drying with $CO_2$, samples were mounted onto aluminum stubs with adhesive tape, sputter coated with gold-palladium, and examined in a Zeiss Merlin field emission scanning electron microscope (Zeiss, Oberkochen, Germany). Images were taken with the SEM software version 5.05 at an acceleration voltage of 5 kV with the Inlens SE-detector and HESE2 SE-detector in a 75:25 ratio. TEM (transmission electron microscopy) was performed as described before (81). To determine OMV size by TEM analysis, in total, 1,421 OMVs in ten fields of view were measured. For size determination, vesicles were negatively stained with 2% aqueous uranyl acetate.

**Fatty acid analysis.** For the fatty acid analysis, samples were prepared from approximately 60 mg *D. shibae* wet cell material or 10 mg *D. shibae* vesicle preparation according to the highly standardized Sherlock microbial identification system (MIS) (MIDI, Microbial ID, Newark, DE, USA). Samples were dried and resolved in 40 $\mu$l *tert*-butylmethylether (MTBE). Following the gas chromatography-flame ionization detection (GC-FID) analysis of the MIDI system, 1 $\mu$l of the sample was injected into an Agilent 7890B gas chromatograph equipped with an Agilent 7000D mass spectrometer (Agilent Technologies, Santa Clara, CA, USA). The injector was set to 170°C and heated to 350°C at 200°C/min and held for 5 min. The GC run started at 170°C, and the program was as follows: 3°C/min to 200°C, 5°C/min to 270, and 120°C to 300°C, hold for 2 min. The MS source temperature was set to 230°C, the electron energy was set to 70 eV, and the mass range was scanned from 40 to 600 *m/z*. The samples from cell materials were analyzed splitless and additionally with a split of 7.5. Data were evaluated using the MassHunter Workstation software (version B.08.00; Agilent Technologies).

**Isolation of vesicle DNA.** OMVs were prepared from 1.5-liter culture as described above. The vesicle pellet was suspended in 2 ml sterile 1$\times$ PBS. To exclude the presence of intact *D. shibae* cells, the sterility of the OMV preparation was checked by spreading 10 $\mu$l of the suspension on an LB and MB agar plate each. After incubation at room temperature for 4 days, no bacterial colonies were found. For each vesicle DNA isolation, 176 $\mu$l of OMV suspension was used. The sample was treated with DNase to remove DNA in the medium or on the vesicle surface. Twenty microliters of 10$\times$ DNase buffer and 4 $\mu$l DNase I (NEB Inc.) were added and incubated at 37°C for 30 min. The enzyme was then inactivated by incubation at 75°C for 10 min. After cooling down for 5 min on ice, disruption of OMVs was performed by adding 2 $\mu$l of 100$\times$ GES lysis buffer (5 M guanidinium thiocyanate, 100 mM EDTA, 0.5% [wt/vol] sarcosyl) and incubating at 37°C for 30 min. RNA was removed by adding 2 $\mu$l of RNase A (20 mg/ml) and incubating at 37°C for 30 min. The samples were then treated with 200 $\mu$l of phenol-chloroform–isoamyl alcohol, vortexed for 1 min, and centrifuged at 12,000 $\times$ *g* for 5 min at 4°C for phase separation. The upper water phase was withdrawn and collected in a new tube (tube 2). Tube 1 was extracted again by addition of 200 $\mu$l of TE buffer (10 mM Tris-HCl, 1 mM disodium salt of EDTA, pH 8.0; Sigma-Aldrich Co.), vigorous mixing for 1 min, and phase separation at 12,000 $\times$ *g* for 5 min at 4°C. The aqueous phase was removed and added to tube 2. The volume of the aqueous phase in tube 2 was measured, and an equal volume of

chloroform-isoamyl alcohol was added. This step was repeated until no protein interphase could be seen (up to 5 times). The aqueous phases were combined and transferred to tube 3. For ethanol precipitation, one-tenth the volume of 3 M sodium acetate (pH 5.2) (Sigma-Aldrich Co.), 1 $\mu$l of glycogen (Thermo Scientific Co.), and 2.5 volumes of cold ($-20°C$) absolute, molecular biology-grade ethanol (Fisher Scientific Co.) were added to tube 3, mixed well, and incubated overnight at $-20°C$. After centrifugation at 12,000 $\times$ g for 5 min at 4°C, the pellet was washed three times with 70% ethanol and centrifuged at 12,000 $\times$ g for 5 min at 4°C. Residual ethanol was removed, the pellet was air dried for 5 min, resuspended in 20 $\mu$l of TE buffer (pH 8.0), and stored at $-70°C$. Three isolations were performed with DNase treatment, and 3 isolations were performed without DNase treatment.

**Sequencing and analysis of vesicle DNA.** Illumina libraries were prepared using the NEBNext Ultra II DNA library prep kit (New England Biolabs, Frankfurt, Germany) according to the manufacturer's protocol. DNA was sheared using a Covaris S220 sonication device (Covaris Inc., MA, USA) with the following settings: 50 s, 105 W, 5% duty factor, 200 cycles of burst. Library preparation was performed according to the protocol. PCR conditions were adapted to the input DNA concentration according to the protocol. Three hundred base-pair paired-end sequencing of the libraries was performed on the Illumina MiSeq system using the v3 chemistry (600 cycles) and according to the standard protocol. Quality trimming of raw reads was conducted with sickle v.1.33 (82). Parameters used for sickle were as follows: paired end sequence trimming, sickle pe; quality value option, –t sanger. Processing and analysis of sequencing data were performed as described before (24). Briefly, reads were mapped to the genome of *D. shibae* DSM 16493$^T$ using Bowtie 2 (83). Discordantly mapping read pairs were discarded. The resulting sam files were converted to indexed binary and pile-up format using SAMtools (84). Accession numbers for the reference genome (1 chromosome, 5 plasmids) are NC_009952.1, NC_009955.1, NC_009956.1, NC_009957.1, NC_009958.1, and NC_009959.1. Pile-up files were loaded into the R statistical environment, and the average read coverage was calculated for sliding windows of 500 nucleotides (nt) for each replicon using the R package zoo (85). Visualization of replicon coverage by vesicle DNA as well as loess regression analysis was performed using the R package ggplot2 (86).

**Preparation of *D. shibae* cells and vesicles for proteome analysis.** Six liters of *D. shibae* culture (4 $\times$ 1.5-liters) were fractionated for one proteome sample each of vesicles and cells. The 6 liters of culture were centrifuged (10,900 $\times$ g for 15 min at 4°C). The complete supernatant was used for the concentration of vesicles according to the protocol described above. The cell pellet from 1 liter of culture was used for the proteome of the cells (the remaining cell pellets were discarded). It was resuspended in 40 ml ice-cold Tris-buffered saline (50 mM Tris-HCl, 15 mM NaCl, pH 8.0), of which, 20 ml was used for the preparation of soluble proteins and the other 20 ml for the preparation of membrane proteins from *D. shibae* cells. Three vesicle preparations were concentrated from the supernatant of 3 $\times$ 6 liters of *D. shibae* culture according to the protocol described above, and three samples for the cell proteome were obtained from the cell pellets of the same batches

**Preparation of membrane and soluble proteins from *D. shibae* cells.** For preparing soluble proteins, cell pellets were washed twice with 1 ml Tris-EDTA buffer (10 mM Tris-HCl, 1 mM EDTA, pH 8.0) and centrifuged (8,000 $\times$ g, 5 min, 4°C). Afterwards, cell disruption was performed by homogenization with glass beads ($\sim$0.1 mm) using the FastPrep instrument (3 $\times$ 3 s, 6.5 m s$^{-1}$; MP Biomedicals). To remove cell debris, cell lysates were centrifuged in two steps: (i) 15,682 $\times$ g for 15 min at 4°C and (ii) 20,879 $\times$ g for 15 min at 4°C. Supernatants of two parallel cultivations were pooled. The supernatant was stored at $-20°C$.

For preparation of membrane proteins from *D. shibae* cells, cells were resuspended in 2 ml lysis buffer (20 mM Tris-HCl, 10 mM MgCl$_2$, 1 mM CaCl$_2$, pH 7.5) after centrifugation and disrupted by homogenization as described above. The obtained cell lysate was sonicated (37 kHz, 1 min, 80 kHz 1 min, 4°C). To remove nucleic acids, cell lysates were incubated with a DNase-RNase mixture (1:100; GE Healthcare) for 40 min at 37°C, and cell debris was subsequently removed by centrifugation (8,000 $\times$ g, 10 min, 4°C).

Separation of membranes from the soluble proteins and preparation of membrane proteins were performed as described previously (87) with the following modification. All ultracentrifugation steps were performed at 100,000 $\times$ g for 1 h and 4°C. Protein samples from parallel cultivations were pooled. Protein concentration was determined with the Roti-Nanoquant (Roth, Karlsruhe, Germany). Protein samples were stored at $-20°C$.

**Preparation of membrane and soluble proteins from *D. shibae* vesicles.** For preparation of proteins from the soluble fraction, the vesicles were resuspended in 200 $\mu$l Tris-EDTA buffer (10 mM Tris-HCl, 1 mM EDTA, pH 8.0) before centrifugation and disrupted by sonification (5 min at 37 kHz, 2 min at 80 kHz, 4°C). After ultracentrifugation (100,000 $\times$ g, 1 h, 4°C), the supernatant containing the soluble proteins was stored at $-20°C$.

To prepare proteins from the membrane fraction of vesicles, the protocol as described for preparation of membrane proteins from cells was used with the following modifications. Before homogenization, vesicles were dissolved in 2 ml ice-cold high-salt buffer; vesicle membrane pellets were resuspended in 50 $\mu$l solubilization buffer followed by reduction, alkylation, and determination of protein concentration.

For GeLC-MS/MS analysis and protein quantification, see Text S1 in the supplemental material.

**Random forest analyses.** riBAQ values from the four fractions were used to train a random forest to predict the localization of the proteins. riBAQ values of proteins that were not found in a fraction were set to zero. Local importance values (49) were derived from the random forest to identify the top 30 proteins that have the highest importance in the random forest for each fraction. The R package randomForest was used to train the random forest and to obtain the local importance values.

**Protein set enrichment analyses.** For protein set enrichment analyses to evaluate whether proteins with specific localizations or functional categories are prevalent in specific samples, all proteins which were identified in at least one of the samples (vesicle membrane proteome, vesicle soluble proteome, cell membrane proteome, and cell soluble proteome) were considered. For each sample and each subcellular localization, a $2 \times 2$ contingency table was generated in which the rows indicated whether a protein belongs to the 700 most abundant proteins and the columns indicated whether a protein belongs to the corresponding subcellular localization. Localizations assigned to proteins were obtained from Locate P v2. The same procedure was performed for functional categories. Functional categories for all proteins were downloaded from the Integrated Microbial Genomes (IMG) database (https://img.jgi.doe.gov/). The proportion of abundant proteins in a subcellular localization or a functional category was considered to be high when Fisher's exact test yielded a value of less than 0.05 after Bonferroni correction for multiple comparisons.

**Data availability.** The mass spectrometry proteomics data have been deposited to the ProteomeXchange Consortium (http://proteomecentral.proteomexchange.org) via the PRIDE partner repository (88) with the data set identifier PXD014351. Sequence reads were deposited at the European Nucleotide Archive (ENA; https://www.ebi.ac.uk/ena) under accession number PRJEB33294.

## SUPPLEMENTAL MATERIAL

Supplemental material is available online only.
**DATA SET S1**, XLSX file, 1.3 MB.
**DATA SET S2**, XLSX file, 0.9 MB.
**MOVIE S1**, AVI file, 1.6 MB.
**MOVIE S2**, AVI file, 1.8 MB.
**TEXT S1**, DOCX file, 0.1 MB.
**FIG S1**, JPG file, 0.1 MB.
**FIG S2**, PDF file, 1.3 MB.
**FIG S3**, PDF file, 0.5 MB.
**FIG S4**, PDF file, 0.2 MB.

## ACKNOWLEDGMENTS

We thank Gesa Martens for excellent technical assistance. We also thank two anonymous reviewers for their constructive criticism and stimulating suggestions.

This work was supported by the Transregional Collaborative Research Center Roseobacter (Transregio TRR 51) of the Deutsche Forschungsgemeinschaft.

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
