## [Reviewer comments · mSystems]

Dinoroseobacter shibae* outer membrane vesicles are enriched for the chromosome dimer resolution site *dif

Hui Wang, Nicole Beier, Christian Bödeker, Helena Sztajer, Petra Henke, Meina Neumann-Schaal, Johannes Mansky, Manfred Rohde, Joerg Overmann, Jörn Petersen, Frank Klawonn, Martin Kucklick, Susanne Engelmann, Jürgen Tomasch, and Irene Wagner-Dobler

Corresponding Author(s): Irene Wagner-Dobler, Helmholtz Centre for Infection Research

Review Timeline:

Submission Date:	July 23, 2020
Editorial Decision:	September 9, 2020
Revision Received:	December 5, 2020
Accepted:	December 14, 2020

Editor: Janet Jansson

Reviewer(s): Disclosure of reviewer identity is with reference to reviewer comments included in decision letter(s). The following individuals involved in review of your submission have agreed to reveal their identity: Harald R Gruber-Vodicka (Reviewer #3)

Transaction Report:

DOI: <https://doi.org/10.1128/mSystems.00693-20>

mSystems00661-19

Dinoroseobacter shibae* outer membrane vesicles are enriched for the chromosome dimer resolution site *dif

Dear Prof. Irene Wagner-Döbler:

I have received the reviews of your manuscript. While your paper addresses an interesting question, the reviewers stated several concerns about your study and did not recommend publication in mSystems. In particular, both reviewers provide recommendations and suggestions for addition of more details of the methods and statistics used.

As you know, at mSystems we are committed to making rapid final decisions. Because it appears that addressing the reviewers' concerns will require a significant amount of additional work that would delay the ultimate outcome, my decision at this time is to reject the manuscript. If you feel that you wish to address the criticisms of the reviewers, you may submit a revised manuscript to mSystems as a new submission, which will be assigned a new manuscript number and receipt date.

Please note the previous manuscript number and my name in the cover letter. Provide point-by-point responses to the issues raised by the reviewers in a file named "Response to Reviewers," not in your cover letter. Upload a compare copy of the manuscript (without figures) as a "Marked-Up Manuscript" file. In the response file, specify with page and line numbers where the revisions have been made in the marked-up manuscript.

I am sorry to convey a negative decision on this occasion, but I hope that the enclosed reviews are useful. The ASM Journals program strives for constant improvement in our submission and publication process. Please tell us how we can improve your experience by taking this quick Author Survey.

Sincerely,
Janet Jansson
Editor, mSystems

Reviewer #1 (Comments for the Author):

Wang et al present a very interesting investigation into the outer membrane vesicles (OMVs) produced by the dinoflagellate symbiont *Dinoroseobacter shibae*. The authors characterize a number of properties associated with the production of vesicles by this organism, reporting that *D. shibae* vesicles are preferentially formed at the septum. They report detailed lipidomic and proteomic characterizations of the vesicles, finding that the vesicles contain largely saturated fatty acids and transport a diverse range of proteins. Through sequencing of the DNA within OMVs, the authors report an abundance of DNA from near the chromosome terminus region, specifically centered around the *dif* site. From these findings, they propose that OMV secretion is associated with cell division and that vesicles might play a role in removing small DNA fragments from around the terminus region. Overall I think that this work is exciting and appropriate for the audience of this journal. I also applaud the authors for advancing models concerning the generation of OMVs and their DNA content.

However, while the data are valuable to the field, there are a number of broad claims made in the paper that either need to be more narrowly reworded or else further supported by additional data or analysis of the existing datasets. The overall approaches are sound, but the ability to interpret these data would also be improved by the addition of some controls, noted below. These may have been done already, but are not described in a way that I could locate. The proposed connection between cell division/membrane remodeling and vesicle generation is certainly reasonable, but the broad claims made about OMV release being tied to cell division are not convincingly supported from the few examples provided here.

Overall, how do you conclude that the 'majority' of vesicles (line 585) emerge from the cell division plane?

Response: We cannot conclude that the `majority` emerge from the cell division plane, you are right. We removed this statement. Instead, we formulate that the vesicles that we isolated from the culture supernatant showed a DNA and proteome signature which is consistent with their secretion during cell division.

Is there more population-level data available?

Response: We do not have more population-level data available. However, we have provided additional TEM and SEM images to show vesicle biogenesis and morphology more clearly.

How do the authors reconcile the conclusion that OMV synthesis is mostly centered around the cell division pole with Fig 1A, showing vesicles all over the cells (did these perhaps just settle back onto the cells?)

Response: The SEM pictures (Figure S4 in the revised manuscript) show that some cells have vesicles of different sizes all over the cell, while others produce none. TEM pictures (Figure 1C, Figure S2, Figure S3 in the revised manuscript) show small and large vesicles emerging from various regions of the cell. The division plane is not visible in those pictures. Only in the time-

lapse movies (Figure 1D-E) one can clearly see that large vesicles emerge from the division plane.

We have removed all statements regarding the “majority” of vesicles. All our data (except for some of the microscopic images) are based on the vesicles shown in the TEM picture of the band that was obtained by ultracentrifugation (Fig. 1A in the revised manuscript). Such preparations were used for the analyses of transcriptome, proteome and fatty acids.

Also in Fig 1B, can you tell from the uncropped image whether the pole shown with tethered vesicles is one which recently arose from a division event? TEM images from other bacteria frequently show the generation of OMVs away from the cell division pole - Fig 2 in Zarantonello et al., "The Cyanobacterium *Cylindrospermopsis raciborskii* (CYRF-01) Responds to Environmental Stresses with Increased Vesiculation Detected at Single-Cell Resolution." *Front Microbiol* 2018; 9: 214-12. comes to mind as one example.

Response: We have replaced Figure 1B with another image which shows the tethering of the vesicle to the cell more clearly (Figure 1C in the revised manuscript). We also provide additional SEM and TEM pictures (Figure S2, S3 and S4). These images show that only some cells in the population produce vesicles (Figure S4), and that vesicles of different sizes are produced by the same cell. They are all derived from the outer membrane and have a single membrane, not a double membrane (Figure S2, S3). It is not possible to observe the division plane in those images.

Regarding Zarantonello *et al.* it is important to keep in mind that our cells are growing happily in a defined medium and do – as far as we can tell – not encounter any type of stress. We have emphasized this in the manuscript.

Are you arguing that cell division-proximal vesicle production is a specific feature of this organism (or Alphaproteobacteria)?

Response: The mechanism of cell division is the most conserved feature in bacteria, and the *dif* site and the XerC/XerD proteins are conserved even across Firmicutes and Proteobacteria. The model is derived from *E. coli* and Alphaproteobacteria. We think that cell-division proximal vesicle production is a feature conserved at least in Proteobacteria. Since at this point we have no data to support this, we have made it very clear that this is a hypothesis.

The proteome data do show some correlations, but is not itself enough to prove that most vesicles arise from cell division. Vesicles visible by phase contrast (shown in Fig 1C and 1D) would likely only be the largest ones compared to the size distribution shown in Fig 2B, so perhaps the largest subpopulation of vesicles might arise from the division plane, while others (the numerically greatest proportion from your data) could come from elsewhere on the cell surface?

Response: We have removed all statements regarding “most” vesicles. However, the DNA, proteome, electron microscopy and fatty acid data are derived from the same method of

preparation by ultracentrifugation. The light microscopic images come from samples that were simply centrifuged down.

We think that the vesicles that were concentrated from the supernatant by ultracentrifugation were derived from the division plane. The small vesicles that can be seen all over the body of some cells might have a different biogenesis. Our data do not allow to decide if those small vesicles contained DNA.

It was also hard for me to tell from the few cells depicted in Movie S1 and Movie S2 that cell division and growth were in fact halted during the period of OMV formation, particularly in comparison to cells that didn't release them. Do you have some quantitative analysis of cell elongation rates from the time-lapse movies which might be able to support this conclusion?

Response: We do not have quantitative analyses of cell elongation rates. Those movies are snapshots and they were striking because vesicles could actually be observed appearing one after the other at the division plane.

Also the possible role of being on an agar pad influencing thrent attachment to the cell should be acknowledged - could the difference between 1C and 1D just involve the local hydration level around those cells on the pad? Was there a difference between cells that underwent rapid separation vs those that seem to stay chained as in S1? That would be an interesting finding that might help support/refine your proposed model for formation.

Response: The formation of vesicles during cell division could be captured only a few times. However, we also observed it in a flow chamber (data not shown) and thus think that it is likely not caused by drying out of the slide.

We have stated that cell division appeared to be halted during vesicle formation in the example shown here (Figure 1D, Supplementary Movie S1). The time lapse movies in the supplement are snapshots that are in accordance with our findings, but do not prove them.

Given that this strain encodes a GTA, which is of similar size to a vesicle, I am also curious about whether the separation procedures used separated vesicles from GTAs, which could complicate the data in Fig 2 and elsewhere. Could you please elaborate on how you ensured that your vesicle preps contained only pure vesicles? I see at least one GTA protein (Dshi_2174 - putative phage major capsid protein) present in the vesicle proteome results.

Response: We think that contamination by GTAs is extremely unlikely for a number of reasons:

- We carefully searched the electron micrographs for GTA particles and couldn't find any.
- The major capsid protein makes up only 0.003% and 0.0015% of the vesicle membrane and soluble fraction, respectively. Thus, it is present only in a negligible amount.
- The packaged DNA speaks against GTA contamination. The terminus region, highly overrepresented in vesicles, is not packaged at all into GTAs.

- Although the *D. shibae* wild-type strain studied here encodes a GTA, it doesn't produce detectable amounts of GTAs. This has been published before [1]. The *D. shibae* quorum sensing mutant $\Delta luxI_2$ however does produce GTAs which could be observed by electron microscopy and Western blotting [1].

Did the authors check the particle content of the media used in the Nanosight to ensure that their particle counts were not influenced by background counts? The Nanosight detection settings seem somewhat high for the camera level and detection limit. As smaller vesicles may likely be dimmer, this could perhaps have contributed at least somewhat to the difference in TEM vs Nanosight sizing.

Response: We measured the particle concentration of the buffer used to resuspend the OMVs after the gradient centrifugation. The buffer contained $2.31 \times 10^6 \pm 7.63 \times 10^5$ particles/ml with an average size of 71.7 ± 4.4 nm. The particle count in the buffer was therefore considered negligible compared to the 10^9 to 10^{10} particles/ml in the vesicle preparation. We have clarified that in the methods.

Relatedly, I found the difference between Fig 2A and 2D striking. To be visible under phase contrast, the vesicles in 2D must have been quite densely clumped. Were such dark clumps at all visible by TEM? Or could this be an artifact of different sample processing methods?

Response: The vesicles in Fig 2A and 2D were prepared by different methods. For staining with fluorescent dyes and counting under the microscope we needed highly concentrated vesicles which were prepared by repeated centrifugation and resuspension steps. Therefore, some clumps appeared. The samples for TEM were treated differently.

However, in the revised manuscript we show staining with a different dye, FM1-43, which emits light when inserted into a membrane (Figure 1F-H). It was used to quantify the fraction of cells additionally stained by DAPI and thus containing DNA inside of their lumen.

Do you have any idea what the brown color mentioned in the methods arises from?

Response: The two bands that appeared after ultracentrifugation were analysed by transmission electron microscopy (see Figure below). The brownish band, used for subsequent analyses, contained pure vesicles. The white band contained mainly flagella.

We speculated that the brown color could have arisen from iron, because we found outer membrane proteins related to iron metabolism in the proteome. Dshi_0570 (ABC-type cobalamin/Fe³⁺-siderophore transport) comprises 0.829% of the membrane fraction of the vesicles and thus belongs to the top 30 vesicle membrane proteins. However, we did not investigate this further. Since the cells are pigmented by chlorophyll and spheroidenone, degradation products of these compounds could also have caused the brown color.

Also, if you are indeed looking at clumps of vesicles in 2D-G, the "65% of vesicles carrying DNA number" could be hugely inflated - it might only take one vesicle in a clump with sufficient DNA to stain by DAPI, whereas many other vesicles could be DNA-free (or at least below staining detection limits) in that same clump.

Response: The number could also be much too small, because small amounts of DNA contained in small vesicles might be below the detection limit. We have clarified that our counts, although they were carefully done, provide only an estimate.

The finding of increased dif sequences in the vesicle DNA is quite exciting, and the fact that this has been seen in other systems speaks to this being a finding of potentially broader importance. However, I would have appreciated a control experiment to ensure that there is no technical reason from the amplification steps involved in the library preparation which could account for the differential abundance. Perhaps this could be done by sequencing cellular genomic DNA from a stationary phase culture, where presumably there should be only one dif site per fully replicated chromosome. If such data are not available, then a more tentative conclusion seems perhaps warranted.

Response: The question, if library preparation results in unequal coverage or selective enrichment of certain regions was already of relevance for our recent publication on packaging of DNA into GTAs for which control sequencing of genomic DNA was performed [1].

We found an even sequencing coverage for DNA isolated from stationary phase cells in contrast to uneven coverage for DNA isolated from GTAs. This is also in accordance with experience from our other sequencing projects where selective amplification during library preparation was never observed. Based on this experience we did not find it necessary to sequence again genomic DNA from cells. We now made it clear in the results section that our findings are in stark contrast to data obtained from genomic DNA sequencing.

Below you can see Figure 3A from [1] to demonstrate coverage of genomic DNA sequencing. Genomic DNA is highlighted in red, whereas GTA DNA coverage is shown in black and the minimum of the cumulative GC/AT-skew (yellow) indicates the terminus of replication region:

A few aspects of the final model would benefit from some additional clarification. If small dif-containing fragments are generated as you propose, how might they be directed into vesicles as opposed to just diffusing away into other parts of the cytoplasm to be turned over? Are you saying that this is a function of the enzymatic steps involved with FtsK/XerCD? Otherwise I'm unclear why relatively small fragments of free DNA would stay constrained to the division site. Also, are you saying that the dif fragments themselves might be contributing to nucleoid occlusion (it seems like that would take a lot of them), or just that the time it takes to resolve these and export them to vesicles indirectly affects nucleoid occlusion by the main chromosome? Could you also clarify in your model exactly how you propose that the dif duplication could occur? Recombination between sister chromatids could certainly lead to duplications of dif, and maybe I'm missing something, but then wouldn't resolving that leave one chromosome without a copy of dif? How might this occur so frequently to show up as a peak in your vesicle data?

Response: We have taken your questions and concerns into account when describing the model in more detail in the revised version. We have replaced Figure 5 with a new Figure (Figure 3 in the revised manuscript) that schematically shows our hypothesis in detail: (A) The localization of *ori*, *ter* and the *dif* sequence on the chromosome. (B) The constructed plasmids that contained tandem *dif* sites and were used to quantify XerC/XerD recombination frequency, (C) the hypothetical generation of tandem *dif* sites by recombination between the two replichores, and (D) the protein machinery at the divisome that contains the XerC/XerD site specific recombinases bound to *dif* and the FtsK translocase. The model also shows the excision of *dif* containing DNA, its release into the periplasm and capture by the outer membrane (D).

Might a model based on duplication from replication slippage be perhaps more parsimonious? And overall the emphasis put on the *dif* site enrichment in this work does not acknowledge the presence of sequence from other regions of the chromosome.

Response: Duplication from replication slippage would not explain our findings. We are seeing enrichment of *dif* plus a region around *ter* which corresponds to the *dif* activity zone determined earlier. Sequences from other regions of the chromosome are present at very low coverage in the vesicles.

Additional minor comments:

37: vesicle not vesicles **Done**

43-47: Please clarify that this is all a proposed model - the role of XerCD, the formation of vesicles prior to septum synthesis, and requirement of vesicle export for completing cell division are suggested, but not proven, by the data shown here.

Response: We have clarified this throughout the manuscript.

51: I might suggest 'domains' instead of 'kingdoms' but up to you **Done**

59: Italicize Salmonella **Done**

76: pH **Done**

78-81: I think this phrasing overstates the conclusions of that paper; reference 15 is careful to say that their data only suggests a model wherein OMVs deliver energy rich compounds to the worm

Response: We inserted “might”

95: content **Done**

103: I would suggest softening the tone of this conclusion; this link is not specifically proven

Response: We have toned down the tone of the conclusion.

115: cultures **Done**

132: was performed **Done**

159: Pelleting vesicles was previously done at 100,000 xg in an ultracentrifuge, and here you washed them twice at 14,000 rpm spinning for only 3 minutes. What was the equivalent xg for the centrifuge used? Might you have lost vesicles / biased the particle count results?

Response: The equivalent g-value for 14,000 rpm of the centrifuge is around 20,000 g. We do not have an ultracentrifuge in our institute for the tiny amount of vesicles that we used for the staining experiment. We used the tabletop Eppendorf centrifuge and found that 14,000 rpm for 3 minutes was sufficient to pellet the vesicles from solution. We have not checked whether

there were still small amounts of vesicles in the supernatant. But for our staining experiment it was important to see whether there is DNAs inside the vesicles rather than to quantify it exactly. Although we still report the number of 65% vesicles with DNA inside, we caution the reader that this value is an estimate.

168: FM1-43 **Done**

205: Given the apparent amount of DNA outside of the vesicles, did you run a control to confirm that the DNase I treatment was sufficiently thorough under these conditions?

Response:

We did not check if remaining DNA might be found in the supernatant after DNase treatment. However, we think DNase treatment was sufficient for the following reasons:

The amount of DNA that we got from the samples before DNase treatment was only 60 ng (Qbit). Treatment should have been far more than enough to eliminate this amount of extravesicular DNA. According to the product information 1 unit of DNase I (neb) completely degrades 1 µg of pBR322 DNA in 10 minutes at 37°C in DNase I Reaction Buffer. We used 8 units and incubated for 30 minutes.

Moreover, we sequenced the control sample that was not treated by DNase. The difference in coverage of chromosome and plasmids with and without DNase treatment (Figure S6,S7) is striking, especially if you look at the boxplots normalized to median coverage of the sample.

252: Membrane **Done**

317-318: Given that your TEM imaging for vesicle size was only done with negative staining, you are not looking within the vesicles to tell whether or not they have two membranes, unless there are other thin sections not shown?

Response: We now provide additional TEM images which clearly demonstrate that the vesicles were enclosed by one membrane only and that they developed from the outer membrane (Figure S2, S3).

323: Might this have resulted in part on the sensitivity and detection settings used for the Nanosight? Smaller particles will typically be less refractive.

Response: We do not think that the setting were the reason. The detection limit of the Nanosight instrument claimed by the company is 10 nm, but a methods comparison of ZetaView and Nanosight NS300 with TEM data showed that “However, both devices failed to report a peak EV diameter below 60 nm compared to TEM” [2].

328-330: From what do you base the assumption that only a subpopulation of cells are producing OMVs in an exponentially growing, steady-state culture? Doesn't this also make the assumption that vesicles are only produced and never removed from the system?

Response: Scanning electron micrographs from exponentially growing cultures showed only very few cells producing vesicles (new Figure S4).

399-401: Please clarify, I don't understand the point about conservation of growth modes when you mention both lateral and polar elongation?

Response: It is out of context and was removed.

421-424: Could you please clarify the relevance of this discussion

Response: The model was developed more clearly, and the paragraph was rewritten.

449: enrichment

Response: Corrected.

549-550: italicize gene names

Response: Corrected.

581: was enriched

Response: Corrected.

Fig 3: the relevance of fitting the distribution of a loess model is not clear

Response: You are right; the message of this figure does not necessarily require the model fit. We removed it. In addition, we noticed that we made two mistakes in the figure legend. Panel A and B show the coverage with 500 nt resolution in 250 nt steps (not 1 kb). Panel C shows the *dif* site with single nucleotide resolution (not 40 nt). This has been corrected.

Fig 5: I thought part C also seemed similar to Fig 1 in Fournes et al, "FtsK translocation permits discrimination between an endogenous and an imported Xer/*dif* recombination complex" PNAS 113:28 (2016). Perhaps this is just coincidental, but if it was indeed a source then it should be cited as well

Response: Of course. There were very many publications studied to start understanding how FtsK-XerC/XerD-*dif* works. Citing this paper was simply forgotten. It is now included.

Fig S1 - is the Y axis number per mL?

Response: Yes, it was added to the y-axis label. Also the standard deviation of the three biological replicates is shown and explained in the legend.

Fig S2 - theoretical

Response: Was corrected.

Fig S3: Comparing genome coverage between studies depends on the sequencing depth/library size for each. It would be easier to interpret these data as relative coverage values among chromosomes/plasmids within each replicate.

Response: Thanks for pointing this out. We added a second panel to this supplementary figure that shows the coverage normalized to the median coverage of the respective sequencing run. This allows a better comparison between the samples.

Please check your supplemental tables (ie Table S5) to remove VLOOKUP links to other spreadsheets.

Response: They were removed.

References: While I did not do an exhaustive check, references 42, 89, and 90 all seem to be incomplete.

Response: They were corrected.

Reviewer #3 (Comments for the Author):

I congratulate the authors on their observations, this was a fascinating read and I largely agree with their conclusions.

Response: Thank you very much!

I have three major concerns:

I generally agree with large parts of the model, and many parts are supported by the data, but they are not well argued in the R&D part. E.g. what is the evidence for duplications/multiplications of the *ter/dif* region that you are drawing in 5C? But more importantly there are also parts in the model I can not follow, e.g. How does the DNA get across the IM and into the periplasm to be packaged? You are suggesting a membrane invagination, of which membrane? The FtsK-XerCD complex sits on the IM, the invagination of the OM to incorporate it together with the excised extrachromosomal *dif* regions would breach cellular integrity. IM proteins are not enriched in the vesicles, so it appears to me that the DNA is exported to the periplasm and then taken up. This needs clarification.

Response: Thank you for these stimulating suggestions. We have incorporated them into the proposed model. Figure 5 (now Figure 3) was completely redrawn to focus basically on the points that you are raising.

We postulate that *dif* was duplicated because otherwise the chromosome would be defect after XerC/XerD recombination. Such duplications have been constructed experimentally and their excision by the XerC/XerD recombinases was used to quantify recombination frequency. These papers are striking, because they mapped the activity of XerC/XerD enzymes to the region around *ter* which corresponds to the enriched genes in our vesicles. We cite these papers and we have added a scheme showing this experimental approach (Figure 3B).

In Figure 3C we schematically show the hypothetical duplication of *dif* by recombination between sister replichoes and the excision of a *dif* containing DNA fragment.

Your second point, how does the DNA fragment which is excised at the cytosolic side of the inner membrane get into the vesicles? Well we can only agree that the DNA must be exported into the periplasm, but we do not know by which enzyme or mechanism. This part of the hypothesis is visualized in Figure 2D.

My second major concern is that the structure of the paper needs substantial improvement. A large part is an inventory style listing of observations with short excursions into the connected literature. I would see a structure that focuses on the model and the presentation of the evidence in close context as a possible way to make the paper more comprehensible and clearer to the reader. You could then move some parts of the inventory that are not directly relevant to the model you are developing to the supplementary text.

Response: We think that this is a very useful suggestion and have followed it. The organization of the paper is now completely changed. We are starting with the microscopic observations,

because this is the first investigation of roseobacter OMVs. Then follows the sequencing of the DNA inside the vesicles. At this point we develop the model based on the key observation of enriched *dif* sequences and the genes around the hypothetical terminus (Figure 3).

We then describe the proteome data. We provide two additional analyses, namely a random forest test (Table S5) and an analysis of unique abundant proteins in the vesicle membrane (Figure 4D). We improved Figure 4 and added a scheme of the most abundant proteins in the vesicle membrane (Figure 4C). Since the cell proteome is already described very briefly, we did not shift any of those results into the supplements.

My third major concern is that the authors do not use the potential of statistical testing or modelling. You have replicated all conditions, I would love to see testing of deviations in the distributions of the proteins from neutral distributions. You could use tools that take the compositionality into account and measure proportionality of gene enrichment, test for significance in enrichment, or use predictive modelling, e.g. to unravel which proteins are linked to which conditions using random forest models.

Response: We have followed your suggestion in two ways: We used a random forest machine learning approach to determine which proteins in our dataset had the highest probability of predicting the localization of a protein among the four investigated proteome fractions (vesicle membrane and soluble proteome, cell membrane and soluble proteome) (Supplementary Table S5). Interestingly, for the vesicle membranes, those proteins were almost the same as the top 30 proteins.

Second, we used a complementary approach to identify unique vesicle membrane proteins. Setting a cumulative riBAQ of 90% as a cut-off we found 460 cell membrane and 225 vesicle membrane proteins. Of those, 65 proteins were unique for the vesicle membrane fraction. Strikingly, FtsK and ZapA were among them. Those are components of the divisome and they are attached to the inner membrane, so finding them inside of outer membrane vesicles of *D. shibae* supports our hypothesis that vesicles are generated during cell division.

Please replace methods centric paragraph headlines or figure headlines with main conclusions for the paragraphs/figures. E.g. Cells appear to be arrested in late division when secreting OMVs. This is mentioned in the legend and could form the lead sentence of the figure legend, instead of reporting the methods used.

Response: That was done.

I would suggest to add more information to Figure 4, there is lots of detail discussed in the written part that is not represented in the figure. One important item could be larger categories e.g. COG categories of functions that are encoded, to represent the functions of the top30 proteins they discuss in the text. The stacked bar charts are also not easy to compare, violin like bar chart comparisons could add to readability.

Response: We have added violin like bar charts to the figure (Figure 4 in the revised manuscript).

Functional categories were added to Tables 1-4.

And finally, to provide a graphical representation of the findings discussed in the text, we schematically show the most important proteins in the vesicle membrane in Figure 2C.

We additionally show the analysis of the membrane proteins comprising 90% of the total riBAQ (Figure 4D).

I do not understand figure 5B. The arrangement is not in agreement with the sketch of chromosomal arrangement in 5A and the z-ring is in a strange position. The distribution of the most abundant proteins is also not adding to clarity. Why is the replicon on the left side and not somehow constricted by the z-ring.

Response: The figure was removed. The model is now shown in Figure 3.

Genes in OMVs: Could you give a full table of all genes and their coverages and not only the filtered lists for >100x coverage and >40x coverage. What about intergenic regions Could you provide a map of the whole chromosome with piled up coverages of the reads in the OMVs

Response: We added two supplementary tables. The first one contains the mean coverage for each gene and the intergenic regions (Table S3A). We also calculated the Pearson correlation between gene and intergenic coverage. The high value of 0.91 indicates that there was no discrepancy between both coverages. The second one (Table S3B) contains the coverage for the sliding windows used for Figure 2A and B. We did not incorporate the coverage on a single nucleotide resolution as this table would be too large for excel.

Minor comments

L71 Vibrios is colloquial, please be more formal and give taxonomic units - Vibrio spp. or members of vibrionaceae

Response: Corrected.

L132 was performed - blank is missing

Response: Corrected.

L138 add a paragraph for 'To follow OMV ...'

Response: Was done.

L140 what do you consider as biological replicates - aliquots of the same inoculation culture?

Response: Three separate 100 ml cultures were set up, and aliquots of those three biological replicates were measured each time. We have made it clearer.

L236 which parameters were used with sickle?

Response: They were added: paired end sequence trimming: sickle pe, quality value option: -t sanger

L296 Fig1A only shows a single cell with relatively large OMVs, could you provide more images in the supplementary and show at least the two extreme cases instead of the current figure 1a

Response: This is a misunderstanding. Fig1A shows six cells of *D. shibae* with different sizes. This heterogeneity in cell size is a feature of *D. shibae* and it is controlled by quorum sensing [3]. One of the smaller cells is in the process of budding, another frequent observation of *D. shibae*. wild-type cultures.

The large cell in the center is covered with tiny OMVs, while the smaller cells have different amounts of OMVs on the surface, some of them slightly larger.

We have shifted this figure to the supplement, because it is apparently misleading. Moreover, we have added additional SEM pictures to the supplementary information showing the heterogeneity of OMV secretion in the population and the size of the vesicles that appear on the cell surface (Figure S4).

L300ff Fig1B only gives a small section of the cell, it is not possible to discern the location on the cell. Please show the full cell and then a e.g. 10x magnified OMV in a separate panel. Please provide more images in the supplementary to support the results.

Response: We have replaced it with a better image. We also provide additional TEM and SEM images (Supplementary Figures S2, S3, S4).

L320 The box plot with dots in 2C does not allow to see the distribution, like in 2B. You could provide Fig. 2B and 2C as a violin plot with TEM on one side and Nanosight on the other.

Response: The continuous distribution across the whole size range shown in Fig. 2B for the Nanosight data cannot be provided for the TEM data. A violin plot of Figure 2C would not provide significantly more information than the box plot with dots that is currently used. We show it below for your information but have left the figure unchanged.

L425 is there 'and are' missing between [...] growth, required [...]

Response: Indeed. Thank you.

L432ff could you give enrichment factors or reduction factors? Calculating and comparing them from % values is tedious.

Response: Has been done.

L476 remove '.' after underrepresented

Response: Was done.

L479 add a comma after vesicles

Response: Was done.

Chapter 'Protein inventory': enrichment factors would also be useful here to identify the most enriched proteins

Response: Determining protein enrichment factors or induction ratios based on riBAQ values is not meaningful because always two different types of samples are being compared, e.g. cellular membranes (comprised of inner and outer membrane) with vesicle membranes (comprised of outer membrane only). Similarly, the soluble fraction of whole cells, comprised of cytoplasm and periplasm, is compared to the soluble fraction of vesicles, comprised of periplasmic proteins. By calculating enrichment factors we would overestimate or underestimate a lot of proteins.

There is no reference in the literature for testing deviation in the distribution of the proteins.

As we have seen in our analyses the composition of the membrane protein extracts of cells and vesicles as well as the composition of soluble protein extracts of cells and vesicles is completely different. Therefore we ranked the relative abundance for each protein in the respective protein extracts and looked in more detail at the top 30 proteins from each extract. Two additional analyses have been added in the revised version (see above).

L521 - 522 This sentence needs commas after the and as well as after proteins.

Response: Was split into two sentences.

L523 the last sentence is self-evident from the lines above and can be removed

Response: Was done.

References

1. Tomasch J, H Wang, ATK Hall, D Patzelt, M Preusse, J Petersen, H Brinkmann, B Bunk, S Bhujju, M Jarek, R Geffers, AS Lang, I Wagner-Döbler. Packaging of *Dinoroseobacter shibae* DNA into Gene Transfer Agent Particles Is Not Random. *Genome Biol Evol* 2018; **10**: 359–369.
2. Bachurski D, M Schuldner, PH Nguyen, A Malz, KS Reiners, PC Grenzi, F Babatz, AC Schauss, HP Hansen, M Hallek, E Pogge von Strandmann. Extracellular vesicle measurements with nanoparticle tracking analysis—An accuracy and repeatability comparison between NanoSight NS300 and ZetaView. *J Extracell Vesicles* 2019; **8**.
3. Patzelt D, H Wang, I Buchholz, M Rohde, L Gröbe, S Pradella, A Neumann, S Schulz, S Heyber, K Münch, R Münch, D Jahn, I Wagner-Döbler, J Tomasch. You are what you talk: quorum sensing induces individual morphologies and cell division modes in *Dinoroseobacter shibae*. *ISME J* 2013; **7**: 2274–2286.

September 9, 2020

Prof. Irene Wagner-Döbler
Technical University of Braunschweig
Microbiology
Braunschweig 38106
Germany

Re: mSystems00693-20 (*Dinoroseobacter shibae* outer membrane vesicles are enriched for the chromosome dimer resolution site *dif*)

Dear Prof. Irene Wagner-Döbler:

Thank you for submission of the revised version of your manuscript. This version is much improved and after minor modification should be acceptable for publication in mSystems. Please note that although both reviewers found that the model was exciting, they indicate that it needs more explanation in the results and a better figure to clarify the model and hypothesis. The reviewers provide suggestions in their comments. In addition, please reduce discussion of the proteomics data to focus on the key findings. One of the reviewers noted that the methods for the random forest analysis of the proteomics data were not presented and it was therefore difficult to judge whether correct statistical approaches were applied. Please make sure to add this information to the revision.

Below you will find the comments of the reviewers.

To submit your modified manuscript, log onto the eJP submission site at <https://msystems.msubmit.net/cgi-bin/main.plex>. If you cannot remember your password, click the "Can't remember your password?" link and follow the instructions on the screen. Go to Author Tasks and click the appropriate manuscript title to begin the resubmission process. The information that you entered when you first submitted the paper will be displayed. Please update the information as necessary. Provide (1) point-by-point responses to the issues raised by the reviewers as file type "Response to Reviewers," not in your cover letter, and (2) a PDF file that indicates the changes from the original submission (by highlighting or underlining the changes) as file type "Marked Up Manuscript - For Review Only."

Due to the SARS-CoV-2 pandemic, our typical 60 day deadline for revisions will not be applied. I hope that you will be able to submit a revised manuscript soon, but want to reassure you that the journal will be flexible in terms of timing, particularly if experimental revisions are needed. When you are ready to resubmit, please know that our staff and Editors are working remotely and handling submissions without delay. If you do not wish to modify the manuscript and prefer to submit it to another journal, please notify me of your decision immediately so that the manuscript may be formally withdrawn from consideration by mSystems.

Sincerely,

Janet Jansson

Editor, mSystems

Journals Department
Reviewer comments:

Reviewer #1 (Comments for the Author):

Wang et al have carried out an extensive revision of their previously submitted manuscript, and have responded in detail to the questions and concerns raised. Previously raised methodology questions have been satisfactorily addressed. The subject matter is appropriate for the journal, I remain excited by their finding of enrichment of dif-containing sequences within the vesicle population. I applaud the efforts that the authors put into addressing the very challenging problem of understanding how DNA is exported into vesicles. There is also quite a lot of very nice microscopy work in here. The authors present an exciting mystery which sets the stage for much followup work.

Unfortunately, I found the revised presentation of the model for dif enrichment to be quite confusing, as described either in the conclusion or Fig. 3 legend. The details of the model appear to be largely modeled after the experimental setup for mapping the dif-active region as shown in Figure 3B. I would suggest that 3B doesn't really help the reader to understand the model, and you'd be better off drawing out more of the intermediate steps in this excision in more detail, or at least to the degree possible.

I am unclear about many aspects of Figure 3C. As I interpret the top part, you appear to have two double stranded molecules from the left and right replichores, each of which ends as a linear molecule with a free end? Was this your intention? Secondly, the coloring of the blue/yellow boxes in the dif site indicates that an unexplained inversion has occurred in the dif site on the right replichore relative to gene 1. Homologous recombination between the two molecules, crossing over within region 1 as shown (is this a single or double crossover?), would also not seem to lead to the intermediate structure shown. I think that in Fig 3A and 3C, more clearly drawing/coloring both the parent and daughter strands of both replichores to better depict the semiconservative replication of the chromosome, and the nature of the molecules remaining at the end of replication, then following your proposed excision, would be helpful, as would more detail of the excision steps. How

this works is all clearly a mystery, but the model as presented does not really shed light on the problem, and needs to consider various options for how the extra copy of dif is generated by the replication machinery in the first place.

Concerning how the excised dif region would get into the periplasm, you may want to examine Liao et al 2014, JBact (doi:10.1128/JB.01493-14), who suggest that the protein secretion machinery could participate in DNA export into vesicles in *S. mutans*.

I understand that the random forest prediction (474-484) was carried out in response to a suggestion from another reviewer, but it is unclear to me exactly what is going on here. The question that this is supposed to be addressing is "which proteins were most likely to predict the location of the proteins in membranes and soluble fractions" which was confusingly worded - you're trying to ask which proteins predict the localization of either themselves, or other proteins into each of your 4 proteome samples? This seems an unusual application of random forest prediction, which is usually based on including a variety of parameters about each data point (here, the proteins) in a training set, and examining the relative importance of those parameters to the final classification, not asking about the relative importance of the data points themselves. What parameters were used to build the training classification? Was it just the rBAQ values and then which fraction the protein was found in? You seem to be recapitulating the localization information that you already had from your initial data, or else if I am mistaken then the section needs to be rewritten. Either way, I think a more straightforward way to address that reviewer's original desire for more statistical analysis would be to carry out something along the lines of a gene set enrichment analysis or another related type of analysis which considers the number of genes/proteins present in a given pathway or functional category in the genome, and then asks whether a set of observed genes/proteins are present at a higher frequency than would be expected by random chance (there are ways to do this with GO term enrichment, using MetaCyc tools, etc).

Other comments:

36: space or "-" between dif and XerC/XerD

41: Should "16:00" be C16:0?

45: "...this type of vesicles" - please specify what type you're referring to, and as compared to which other types?

49-50: "We studied OMV cargo.... using single cell analysis": please clarify, the single cell analysis you refer to must be some of the various microscopy, which didn't specifically tell you about cargo

54: chromatids

256: is the loess regression still used in the paper somewhere?

347-354: While Fig S2C is a beautiful SEM image, I am not convinced that this one image of a handful of cells supports the conclusion that 'very few cells in the population produce OMVs.' Many of those cells to my eye have various small bumps that could be emerging vesicles. It is perhaps consistent with heterogeneity of vesicle release among cells.

416: This is not a rule; some bacteria have linear chromosomes. See for instance

[https://doi.org/10.1016/S0960-9822\(02\)00916-8](https://doi.org/10.1016/S0960-9822(02)00916-8)

503-615: As a general comment, this section could be condensed significantly and still get across the main message about the different categories of proteins found in the vesicles.

677: In most of the paper you talk about the Inner Membrane (IM) but use "CM" in this figure label instead (and the abbreviations are not noted)

Reviewer #3 (Comments for the Author):

I congratulate the authors on the much improved manuscript. The new analyses together with the updated model make a convincing case that OMV formation in *Dinoroseobacter shibae* and likely many more bacteria might also be linked to waste removal after somewhat messy DNA replication. I have a couple of suggestions to further improve the manuscript, and to make it even more accessible:

1) To me the figure 3B could go into the supplement, it is an illustration of the recombination activity around *dif* as shown in a previous publication. It is helpful to understand the recombinatory window around *dif*, but is not necessary for the main arguments of the manuscript

2) I would suggest to move 3D into a separate figure 5 and present it in the concluding summary.

3) The concluding summary reads very much like a staccato of all the cool things found, but stays too superficial in the concluding remarks- What are evolutionary consequences - e.g. should we check *dif* and the surrounding of *dif* next time we annotate a genome of an organism that we know produces OMVs?

4)

There are a couple of minor things like typos or misplaced commas that the authors might want to fix

L60 I would write Gram-negative bacteria with a - between Gram and negative. This is the common way and e.g. also used in the cited literature, see the title of citation # 4.

L90 please give a taxonomic background and mention that the Rhodobacteraceae are Alphaproteobacteria

L158 remove comma after agarose-pad. I also think that the - in agarose pad is not necessary.

L178 Please put the details of the DNase treatment up front at the first time you mention it. You can then refer to this 'as above'

L188 I would put a comma before samples

L202 please explain that the Agilent 7890 is a gas chromatograph

L211 OMVs is missing the s

L216-219 The details for the DNase give here should move up to L178

L244 I guess you mean the manufacturer's protocol

L256 which R package did you use for this?

L260 I am confused as to the 1.5 or 1 liter were used.

L278 DNase and RNase were used as before?

L330-336 reads very anecdotal. Could you do some kind of quantification?

L374 this is likely caused by the replication process running in parallel and in a non-synchronized manner across the whole population. As all the cells start replicating around the *ori*, but are in different stages in an unsynced state. You could explain this here.

L448 I would delete the last sentence and instead finish the sentence with something from your own data - e.g. the fact the presence of *dif* across such a broad diversity of bacteria suggests that what we are looking at in terms of OMV secretion is something broadly conserved

L485 the 'top 30 proteins' can be removed.

L540 I would delete 'the' before green

L634 was ALSO found

L645 ... are constitutively expressed ...

mSystems00693-20 (*Dinoroseobacter shibae* outer membrane vesicles are enriched for the chromosome dimer resolution site *dif*)

Reviewer comments:

Reviewer #1 (Comments for the Author):

Wang et al have carried out an extensive revision of their previously submitted manuscript, and have responded in detail to the questions and concerns raised. Previously raised methodology questions have been satisfactorily addressed. The subject matter is appropriate for the journal, I remain excited by their finding of enrichment of *dif*-containing sequences within the vesicle population. I applaud the efforts that the authors put into addressing the very challenging problem of understanding how DNA is exported into vesicles. There is also quite a lot of very nice microscopy work in here. The authors present an exciting mystery which sets the stage for much followup work.

Response: Thank you!

Unfortunately, I found the revised presentation of the model for *dif* enrichment to be quite confusing, as described either in the conclusion or Fig. 3 legend. The details of the model appear to be largely modeled after the experimental setup for mapping the *dif*-active region as shown in Figure 3B. I would suggest that 3B doesn't really help the reader to understand the model, and you'd be better off drawing out more of the intermediate steps in this excision in more detail, or at least to the degree possible.

I am unclear about many aspects of Figure 3C. As I interpret the top part, you appear to have two double stranded molecules from the left and right replichores, each of which ends as a linear molecule with a free end? Was this your intention? Secondly, the coloring of the blue/yellow boxes in the *dif* site indicates that an unexplained inversion has occurred in the *dif* site on the right replichore relative to gene 1. Homologous recombination between the two molecules, crossing over within region 1 as shown (is this a single or double crossover?), would also not seem to lead to the intermediate structure shown. I think that in Fig 3A and 3C, more clearly drawing/coloring both the parent and daughter strands of both replichores to better depict the semiconservative replication of the chromosome, and the nature of the molecules remaining at the end of replication, then following your proposed excision, would be helpful, as would more detail of the excision steps. How this works is all clearly a mystery, but the model as presented does not really shed light on the problem, and needs to consider various options for how the extra copy of *dif* is generated by the replication machinery in the first place.

Response: In fact, we were not satisfied with it ourselves. We now studied the literature on DNA replication, particularly termination, and worked together with a geneticist, Dr. Jörn Petersen, to work out the details of a hypothetical scenario.

We are now suggesting a new model. The basic idea is that there may be physiological situations when the two replisomes do not meet exactly at 180°. The literature suggests that one of them can be delayed because, for example, conflict with transcription. The other replisome then continues beyond 180° and over-replicates some genes. The two replisomes thus collide outside *ter*. The delayed replisome then continues towards *ter*,

but since the original DNA double strand has already been replicated, it uses this copy for a second round of synthesis. This is termed “template switching” in the literature. The delayed replisome also over-replicates some genes in the process and copies *dif* again. The actual situation may be much more complex, with fork reversal and transient over-replication by one or both forks.

We hypothesize that the fragments found in our vesicles result from repair of over-replication and have developed two hypothetical scenarios for repair of over-replication

Their main difference between them is that in scenario (1) repair of over-replicated fragments occurs by endonucleases and in scenario (2) repair occurs by site-specific recombination at *dif* by the XerC/XerD enzymes. It is therefore straightforward to decide between those two possibilities in future experiments.

Concerning how the excised *dif* region would get into the periplasm, you may want to examine Liao et al 2014, JBact (doi:10.1128/JB.01493-14), who suggest that the protein secretion machinery could participate in DNA export into vesicles in *S. mutans*.

Response: We have examined the paper. Their main focus was on eDNA in biofilms. Deletion of sortase A caused a reduction in eDNA both in biofilms and planktonic culture. They detected membrane vesicles (MVs) for the first time in *S. mutans*. Deletion of sortase A had no effect on MV production, but these MVs had less surface-associated proteins. This would be expected since sortase A is a membrane-localized transpeptidase that cleaves and covalently anchors proteins to the peptidoglycan.

The authors did not investigate DNA inside the vesicles. Moreover, *S. mutans* is Gram-positive, and *D. shibae* is Gram-negative. Sortase A is not a universal protein secretion mechanism but specific to Gram-positive bacteria [1].

So we think there is no direct hint in this paper as to the mechanism how DNA can be exported into the periplasm and from there into vesicles.

I understand that the random forest prediction (474-484) was carried out in response to a suggestion from another reviewer, but it is unclear to me exactly what is going on here. The question that this is supposed to be addressing is "which proteins were most likely to predict the location of the proteins in membranes and soluble fractions" which was confusingly worded - you're trying to ask which proteins predict the localization of either themselves, or other proteins into each of your 4 proteome samples? This seems an unusual application of random forest prediction, which is usually based on including a variety of parameters about each data point (here, the proteins) in a training set, and examining the relative importance of those parameters to the final classification, not asking about the relative importance of the data points themselves. What parameters were used to build the training classification? Was it just the riBAQ values and then which fraction the protein was found in? You seem to be recapitulating the localization information that you already had from your initial data, or else if I am mistaken then the section needs to be rewritten.

Response: Indeed, the random forest prediction was applied in an uncommon manner. The random forest was trained to predict the localization of each protein based on the four riBAQ values. The purpose of training the random forest was not the predictions themselves but to identify those proteins that had the highest influence on the random forest in terms of the so-called local importance values (Y. Lin, Y. Jeon (2006). Random

Forests and Adaptive Nearest Neighbors. Journal of the American Statistical Association 101, 578–590) that can be derived from the training of the random forest.

After the random forest was trained, proteins of each sample were sorted by their local importance. Considering the predicted localization of the top 30 proteins, in the sample “membrane fraction of vesicles” 26 were predicted to be outer membrane proteins and in the sample type “soluble fraction of vesicles” all of these were periplasmic and outer membrane proteins. At the same time, in the “membrane fraction of cells” inner membrane proteins and in the “soluble fraction of cells” cytoplasmic proteins are predominant among the top 30 proteins.

These data confirmed our hypothesis that in *D. shibae*, vesicles are derived from the outer membrane and enclose periplasmic proteins.

The methods and results were modified accordingly.

Either way, I think a more straightforward way to address that reviewer's original desire for more statistical analysis would be to carry out something along the lines of a gene set enrichment analysis or another related type of analysis which considers the number of genes/proteins present in a given pathway or functional category in the genome, and then asks whether a set of observed genes/proteins are present at a higher frequency than would be expected by random chance (there are ways to do this with GO term enrichment, using MetaCyc tools, etc).

Response: We now carried out protein set enrichment analyses (PSEA) to evaluate whether proteins with specific localizations or functional categories are prevalent in specific samples. The results are shown in Data Set S2 Sheet 6 and Sheet 7 and described in the Results.

Other comments:

36: space or "-" between dif and XerC/XerD **done**

41: Should "16:00" be C16:0? **done**

45: "...this type of vesicles" - please specify what type you're referring to, and as compared to which other types?

Response: What we meant was, “vesicles produced during exponential growth under optimal conditions”. This was specified.

49-50: "We studied OMV cargo.... using single cell analysis": please clarify, the single cell analysis you refer to must be some of the various microscopy, which didn't specifically tell you about cargo

Response: We are referring to fluorescence microscopy using DNA stains. We counted the fraction of vesicles that contained DNA. We replaced “single cell analysis” by “microscopy”.

54: chromatids **done**

256: is the loess regression still used in the paper somewhere?

Response: It is used for Fig. S3 in the supplements.

347-354: While Fig S2C is a beautiful SEM image, I am not convinced that this one image of a handful of cells supports the conclusion that 'very few cells in the population produce OMVs.'

Many of those cells to my eye have various small bumps that could be emerging vesicles. It is perhaps consistent with heterogeneity of vesicle release among cells.

Response: The picture is representative of many such pictures. But you are right, many cells show small bumps that could be emerging vesicles. However, very few cells actually have already formed clearly visible vesicles, like the ones seen in the close-ups. We have added: "... although many cells have little bumps on their surface which could be emerging vesicles."

416: This is not a rule; some bacteria have linear chromosomes. See for instance [https://doi.org/10.1016/S0960-9822\(02\)00916-8](https://doi.org/10.1016/S0960-9822(02)00916-8)

Response: You are right, some bacteria have linear chromosomes, but it is an exception and not the rule even today, when more than 6000 bacterial genomes are available. The reference that you referred us to [2] describes two strains with linear chromosomes, namely *Agrobacterium tumefaciens* and *Borrelia burgdorferi*, from a list of 10 newly sequenced genomes in 2002. A 2017 review on genome organization [3] reports linear chromosomes in some strains of *Borrelia*, *Agrobacterium* and *Streptomyces*. Also, a not-yet cultivated plant pathogen *Phytoplasma mali* is mentioned. So I do think that it is fair to state that linear chromosomes exist in bacteria, but they are the exception rather than the rule. We have adjusted the statement accordingly.

503-615: As a general comment, this section could be condensed significantly and still get across the main message about the different categories of proteins found in the vesicles.

Response: We have shortened it.

677: In most of the paper you talk about the Inner Membrane (IM) but use "CM" in this figure label instead (and the abbreviations are not noted)

Response: We replaced CM by IM in the figure label and added the abbreviations.

References

1. Nobbs AH, RJ Lamont, HF Jenkinson. Streptococcus Adherence and Colonization. *Microbiol Mol Biol Rev* 2009; **73**: 407–450.
2. Ochman H. Bacterial evolution: Chromosome arithmetic and geometry. *Curr Biol* 2002; **12**: 427–428.
3. Feijoo-Siota L, JLR Rama, A Sánchez-Pérez, TG Villa. Considerations on bacterial nucleoids. *Appl Microbiol Biotechnol* 2017; **101**: 5591–5602.

Reviewer #3 (Comments for the Author):

I congratulate the authors on the much improved manuscript. The new analyses together with the updated model make a convincing case that OMV formation in *Dinoroseobacter shibae* and likely many more bacteria might also be linked to waste removal after somewhat messy DNA replication.

I have a couple of suggestions to further improve the manuscript, and to make it even more accessible:

1) To me the figure 3B could go into the supplement, it is an illustration of the recombination activity around dif as shown in a previous publication. It is helpful to understand the recombinatory window around dif, but is not necessary for the main arguments of the manuscript

Response: Was done.

2) I would suggest to move 3D into a separate figure 5 and present it in the concluding summary.

Response: Was done.

3) The concluding summary reads very much like a staccato of all the cool things found, but stays too superficial in the concluding remarks- What are evolutionary consequences - e.g. should we check dif and the surrounding of dif next time we annotate a genome of an organism that we know produces OMVs?

Response: We elaborated more on the findings in the concluding summary. Thank you for asking for this (it's normally the other way round)!

4)

There are a couple of minor things like typos or misplaced commas that the authors might want to fix

L60 I would write Gram-negative bacteria with a - between Gram and negative. This is the common way and e.g. also used in the cited literature, see the title of citation # 4.

Response: Was done.

L90 please give a taxonomic background and mention that the Rhodobacteraceae are Alphaproteobacteria

Response: Was done.

L158 remove comma after agarose-pad. I also think that the - in agarose pad is not necessary.

Response: Was done.

L178 Please put the details of the DNase treatment up front at the first time you mention it. You can then refer to this 'as above'

Response: We prefer not to do this, because DNase treatment for microscopy and DNase treatment of the vesicle preparation prior to sequencing was performed in a slightly different way. It was also slightly different for the samples that were used for proteome analysis (see below).

L188 I would put a comma before samples. **Response:** Was done.

L202 please explain that the Agilent 7890 is a gas chromatograph. **Response:** Was added.

L211 OMVs is missing the s. **Response:** Was added.

L216-219 The details for the DNase give here should move up to L178

Response: As it is slightly different from the treatment of samples for microscopy, we prefer to leave it as it is (see above).

L244 I guess you mean the manufacturer's protocol. **Response:** Wording was changed accordingly.

L256 which R package did you use for this?

Response: Pile-up files were loaded into the R statistical environment and the average read coverage was calculated for sliding windows of 500 nt for each replicon using the R-package zoo (DOI: 10.18637/jss.v014.i06).

L260 I am confused as to the 1.5 or 1 liter were used.

Response: We used 4 x 1.5 liters of culture for one sample for the proteome analyses. The complete supernatant of the 6 liters was concentrated for the vesicle proteome. For the cell proteome, the pellet from 1 liter of the culture was used. The excess pellets were discarded. Three such batches of 6 liter were processed.

We have clarified this in the Results.

L278 DNase and RNase were used as before?

Response: Was performed as described, i.e. slightly different than before.

L330-336 reads very anecdotal. Could you do some kind of quantification?

Response: Electron microscopic pictures cannot be quantified properly. The examples shown here are representative of many pictures that were taken and analysed.

L374 this is likely caused by the replication process running in parallel and in a non-synchronized manner across the whole population. As all the cells start replicating around the ori, but are in different stages in an unsynced state. You could explain this here.

Response: Please keep in mind that this was not found for the DNA within the cells, but for that in the vesicles. It is an interesting point. However, it is not clear how this could result in the enrichment of genes around the terminus in the vesicles.

L448 I would delete the last sentence and instead finish the sentence with something from your own data - e.g. the fact the presence of dif across such a broad diversity of bacteria suggests that what we are looking at in terms of OMV secretion is something broadly conserved

Response: Thank you. We followed your advice.

L485 the 'top 30 proteins' can be removed. **Response:** Was done.

L540 I would delete 'the' before green

Response: Sentence was deleted to shorten the description of the proteomic results.

L634 was ALSO found. **Response:** Was added.

L645 ... are constitutively expressed ...

Response: Sentence now reads:

OMVs are constitutively produced by *D. shibae* during undisturbed growth...

December 14, 2020

Prof. Irene Wagner-Dobler
Helmholtz Centre for Infection Research
Inhoffenstreet 7
Braunschweig 38124
Germany

Re: mSystems00693-20R1 (*Dinoroseobacter shibae* outer membrane vesicles are enriched for the chromosome dimer resolution site *dif*)

Dear Prof. Irene Wagner-Dobler:

Thank you for taking such care with the revisions of your manuscript. I am pleased to inform you that your manuscript has been accepted, and I am forwarding it to the ASM Journals Department for publication. For your reference, ASM Journals' address is given below. Before it can be scheduled for publication, your manuscript will be checked by the mSystems senior production editor, Ellie Ghatineh, to make sure that all elements meet the technical requirements for publication. She will contact you if anything needs to be revised before copyediting and production can begin. Otherwise, you will be notified when your proofs are ready to be viewed.

Sincerely,

Janet Jansson
Editor, mSystems

Journals Department
Figure S4: Accept
Data Set S2: Accept
Data Set S1: Accept
Movie S1: Accept
Supplementary Methods: Accept
Movie S2: Accept
Figure S1: Accept
Figure S3: Accept
Figure S2: Accept